# Establishing and using a genetic database for resolving identification of fish species in the Sea of Galilee, Israel

Roni Tadmor-Levi[1,2], Tomer Borovski[2†], Evgeniya Marcos-Hadad[2], James Shapiro[3], Gideon Hulata[4], Daniel Golani[1], Lior David[1,2]*

1 National Natural History Collections and Department of Ecology, Evolution and Behavior, The Hebrew University of Jerusalem, Jerusalem, Israel, 2 Department of Animal Sciences, RH Smith Faculty of Agriculture Food and Environment, The Hebrew University of Jerusalem, Rehovot, Israel, 3 Fishery and Aquaculture Unit, Ministry of Agriculture and Rural Development, Beit-Dagan, Israel, 4 Department of Poultry and Aquaculture, Institute of Animal Science, Agricultural Research Organization (ARO), Volcani Center, Rishon LeZion, Israel

† Deceased.
* lior.david@mail.huji.ac.il

**Data Availability Statement:** Sample collection data and barcode sequences were deposited in Barcode Of Life Datasystems (BOLD) under project name LKCOX. Accession numbers LKCOX001-19 -

## Abstract

Freshwaters are a very valuable resource in arid areas, such as Mediterranean countries. Freshwater systems are vulnerable ecological habitats, significantly disturbed globally and especially in arid areas. The Sea of Galilee is the largest surface freshwater body in the Middle East. It is an isolated habitat supporting unique fish populations, including endemic species and populations on the edge of their distribution range. Using the Sea of Galilee for water supply, fishing and recreation has been placing pressure on these fish populations. Therefore, efficient monitoring and effective actions can make a difference in the conservation of these unique fish populations. To set a baseline and develop molecular tools to do so, in this study, DNA barcoding was used to establish a database of molecular species identification based on sequences of *Cytochrome C Oxidase subunit I* gene. DNA barcodes for 22 species were obtained and deposited in Barcode of Life Database. Among these, 12 barcodes for 10 species were new to the database and different from those already there. Barcode sequences were queried against the database and similar barcodes from the same and closely related species were obtained. Disagreements between morphological and molecular species identification were identified for five species, which were further studied by phylogenetic and genetic distances analyses. These analyses suggested the Sea of Galilee contained hybrid fish of some species and other species for which the species definition should be reconsidered. Notably, the cyprinid fish defined as *Garra rufa*, should be considered as *Garra jordanica*. Taken together, along with data supporting reconsideration of species definition, this study sets the basis for further using molecular tools for monitoring fish populations, understanding their ecology, and effectively managing their conservation in this unique and important habitat and in the region.

LKCOX217-19 https://www.boldsystems.org/index.php.

**Funding:** The study was funded by the Chief Scientist of the Israeli Ministry of Agriculture and Rural Development under Grant #186-0001-11 to LD and GH. https://www.moag.gov.il/en/Ministrys%20Units/Chief%20Scientist/Pages/default.aspx. The funders had no role in study design, data collection and analysis, decision to publish, or preparation of the manuscript.

**Competing interests:** The authors have declared that no competing interests exist.

# Introduction

Despite freshwater habitats only comprising of 1% of the Earth's surface water, they support 43% (13,000 species) of the world's fish species [1, 2], rich in biodiversity across all trophic levels [3, 4]. As a result of humans long standing dependence on water, either for direct uses like irrigation and drinking or indirect uses like hydro-electrical energy production and food fishing, many freshwater fish species have become highly endangered through disturbance and diminishing of natural habitats [5–9]. Further human derived pressures on natural fish diversity has also been documented due to introduction and invasion of exotic fish species [10–13]. Conservation measures intended to mitigate the impact of human derived pressures have largely been slow and inadequate, and as a result, populations of many freshwater species have been declining rapidly [5, 6, 8, 14]. Assessment of freshwater fish biodiversity and population status for effective management is therefore a priority, however, can be considered a great challenge [1, 6, 14, 15].

Due to its arid climatic conditions, Israel have been facing growing demands for freshwater. Consequently, its limited freshwater habitats have been under great human pressure. The Sea of Galilee (Lake Kinneret) is the biggest surface freshwater reservoir in the middle east (ca. 170 km$^2$), and due to its crucial role in water supply, it is highly regulated [16–18]. In terms of freshwater fish, approximately 18 native and eight non-native species were documented in the Sea of Galilee [19]. Native species are dominated by cyprinids and cichlids that are of mainly Asian (Mesopotamian) and African origins, respectively [16]. The native cyprinid Kinneret bleak, *Mirogrex terraesanctae* (Steinitz, 1952), which is endemic to the Sea of Galilee, is probably the most abundant species in terms of both numbers and biomass [20]. The native cichlid, Galilee tilapia, St. Peter's fish or Mango tilapia *Sarotherodon galilaeus* (Linnaeus, 1758), which is widespread in central and western Africa [8], is also abundant in terms of numbers and biomass, based on fishery landings, and is considered an important species to local fisheries. Non-native species, which were unintentionally introduced, include the European eel *Anguilla* (Linnaeus, 1759) that hitchhiked with mullet fry caught in coastal estuaries and stocked in the Sea of Galilee [21], the hybrid tilapia *Oreochromis niloticus* (Linnaeus, 1758) x *Oreochromis aureus* (Steindachner, 1864) that escaped from aquaculture ponds [16, 22], the green swordtail *Xiphophorus helleri* (Heckel, 1848) [22] and the Peacock bass *Cichla kelberi* Kullander and Ferreira, 2006, probably released into the Sea of Galilee by hobbyists [23]. The non-native invasive mosquito fish *Gambusia affinis* (Baird and Girard, 1853) was intentionally introduced to control mosquitoes [16]. Additionally, significant non-native species including the common carp *Cyprinus carpio* Linnaeus, 1758, the silver carp *Hypophthalmichthys molitrix* (Valenciennes, 1844), the thinlip grey mullet *Chelon ramada* (Risso, 1827) and the flathead grey mullet *Mugil cephalus* Linnaeus, 1758 were introduced to enhance commercial fishing [16, 22].

The dynamics of pelagic fish populations in Sea of Galilee is monitored mainly by means of acoustic surveys that measure abundance and distribution of fish shoals and estimate size of fish based on the echo target strength [24]. These acoustic surveys do not distinguish between species, but some inferences can be made based on estimated size of individuals and shoals. Based on these surveys, some estimates for population size of main pelagic species like the *M. terraesanctae* can be obtained [19]. Other inference on the population size comes from the commercial landing of sized fish [20, 25].

In recent years, DNA polymorphism, a more reliable and accurate method, is increasingly used for taxonomic and ecological research [e.g. 26–28]. DNA sequences of specific genes are used successfully to identify biological species in a method called DNA barcoding [29–31]. As certain DNA sequences in genomes evolve at rates similar to evolution of new species, DNA barcoding can complement and enhance species identification. DNA barcoding based

methods can also be utilized for efficient monitoring of freshwater fish biodiversity, as DNA can easily be extracted from various types of samples including from environmental water samples, known as E-barcoding [32, 33]. Barcode sequences obtained from unidentified samples can be matched to available records from known samples for species identification. Thus, the application of DNA barcoding for reliable molecular identification of species, requires a reference database that connects DNA sequences to specific species [34, 35]. The Barcode of life Data systems (BOLD) is a comprehensive, parameterized DNA barcode reference library [36], created to support the use of barcode data and molecular species identification for studying various aspects of biodiversity.

For this study, we applied DNA barcoding to samples taken mainly from the Sea of Galilee to create a reference barcoding database, within the platform of BOLD, for molecular identification of fish in the context of the Sea of Galilee and the region, an important tool for management and conservation of fish populations.

## Methods

### Sampling of fish species

To genetically barcode fish species from the Sea of Galilee and surrounding areas, an extensive sampling scheme was carried out to obtain samples from each of the species that could be found in the area. The Sea of Galilee, the main sampling site, and surrounding sites such as Asi stream (one of Beit She'an streams, south to the Sea of Galilee and part of the eastern watershed of Israel), Ein Afek pools (close by Acre, i.e. Akko, part of the western watershed of Israel) and Ein Te'o spring (close by Hula Lake, northern to Sea of Galilee and part of the eastern watershed of Israel) (see Fig 1A by [27]), were sampled throughout 2013, 2014 and 2015 to collect samples of species that predominantly occur in this region of Israel by means of seine netting. Number of individual samples collected for each site and species are recorded in Table 1. Fish were anesthetized with 200 μL/L 2-phenoxy-ethanol and after taxonomically identified to the species level in the field, a fin clip was taken from every fish. The fins were stored in labeled 2 mL Eppendorf tubes in 99.9% ethanol at -20˚C. In most cases, after identification and sampling, fish recovered from anesthesia and were returned to their environment after a monitoring period of approximately ten minutes. Some fish, however, were sacrificed, photographed and fixed in formalin to be deposited in the National Natural History Collections of the Hebrew University of Jerusalem, Israel with a museum number (Table 1), serving as voucher specimens (transferred to 99.9% ethanol).

### DNA extraction, polymerase chain reaction (PCR), purification and sequencing

DNA from stored fin samples was extracted using a modified protein salting-out method [37]. DNA concentration and quality (Optical Density $OD_{260}/OD_{280}$ ratio) were measured using NanoDrop ND-1000 (NanoDrop Technologies) and visually examined by 1.5% TBE Agarose gel electrophoresis. DNA samples were stored at −20˚C until used for further analysis.

A partial fragment of approximately 700bp of the *Cytochrome C oxidase subunit I* (COI), a mitochondrially encoded gene, was amplified by PCR using primers adopted from [31] (FishF1: 5'-TCAACCACCCACAAAGACATTGGCAC-3' and FishR1: 5'-TAGACTTCTGGGTGGCCAAAGAATCA-3' ). As DNA template, 50 ng of purified DNA were used in a total reaction volume of 20 μL. PCR mix contained: 2 μL 10× PCR buffer, $MgCl_2$ (25mM), dNTPs (6.6mM of each), Taq polymerase (1.5 units), primer mix (forward + reverse, 2.5 μM of each) and 10 μL water to complete the volume. PCR thermal protocol included an

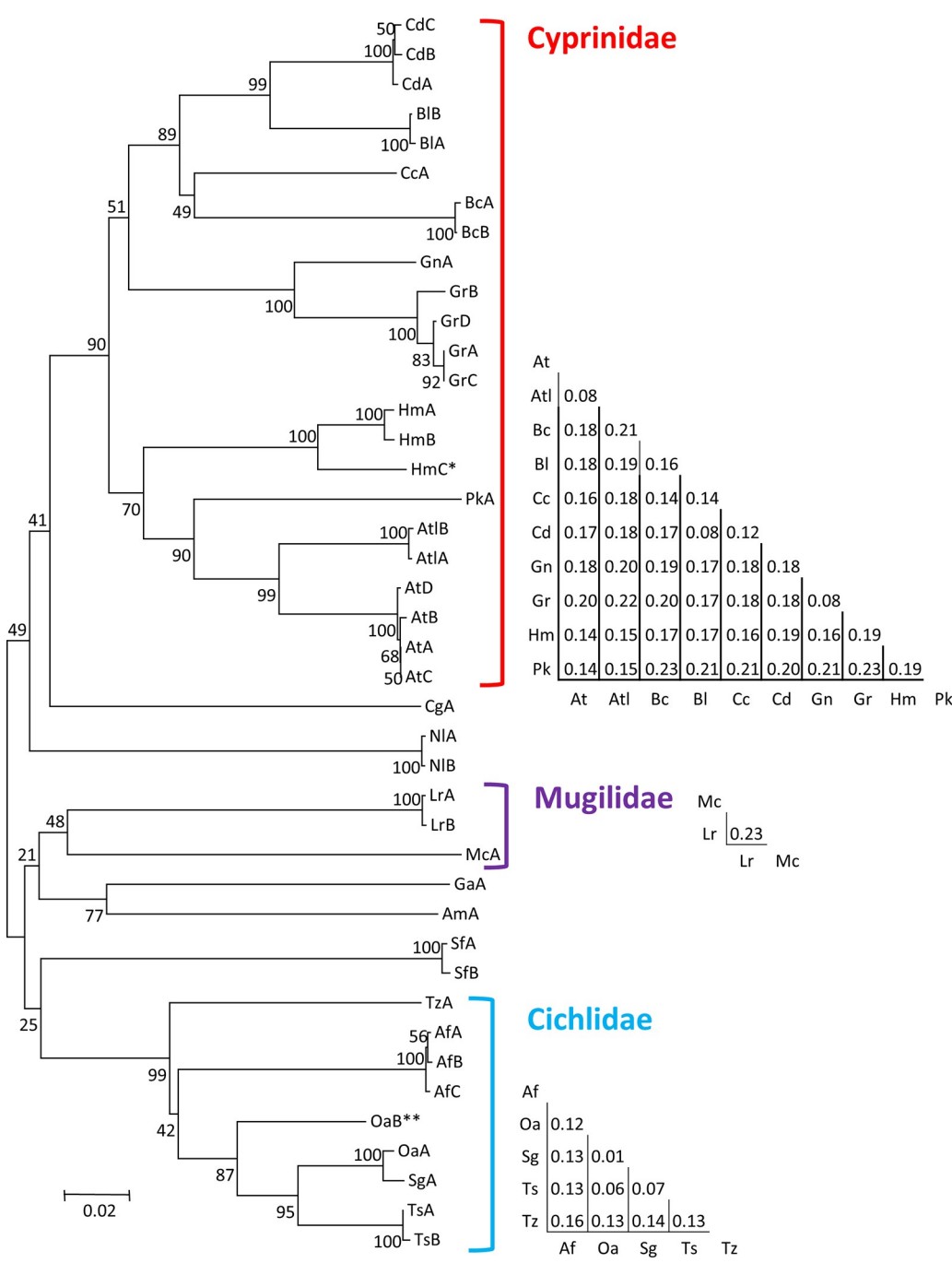

**Fig 1. A neighbor-joining unrooted tree of species based on COI haplotypes.** Next to branch edges are haplotype acronyms as listed in Table 2. Numbers next to bifurcations denote percent bootstrap support. For some species, more than one haplotype was identified (denoted A, B etc.). For major families, which are represented by multiple species, a mean K2P between species distance matrix is given to the right. (*)–HmC haplotype derived from specimens suspected as *Hypophthalmichthys molitrix* x *Hypophthalmichthys nobilis* hybrids, matches *H. nobilis* sequences from BOLD. (**)— OaB haplotype derived from a specimen suspected as *Oreochromis niloticus* x *Oreochromis aureus* hybrid, matches *O. niloticus* sequences from BOLD.

initial step of 94°C for 2 min; followed by a touchdown profile of 94°C for 30 s, annealing from 60 to 53°C for 1 min with a decrease of 0.5°C per each of 14 cycles, and extension at 72°C

**Table 1. Fish species collected, barcoded and deposited at the National Natural History Collections of the Hebrew University of Jerusalem (NNHC-HUJ) by species and location.**

| Order | Family | Species | Sampling location | N samples collected | N successfully barcoded samples | N Specimens deposited | Museum numbers at NNHC-HUJ |
|---|---|---|---|---|---|---|---|
| Siluriformes | Clariidae | *Clarias gariepinus* | Sea of Galilee | 8 | 5 | 0 | |
| Cypriniformes | Cyprinidae | *Mirogrex terraesanctae* | Sea of Galilee | 10 | 8 | 0 | |
| | | *Luciobarbus longiceps* | Sea of Galilee | 10 | 7 | 0 | |
| | | *Acanthobrama telavivensis+* | Ein Afek | 13 | 7 | 3 | HUJ21008 |
| | | *Capoeta damascina* | Sea of Galilee | 5 | 5 | 0 | |
| | | *Cyprinus carpio* | Sea of Galilee | 14 | 5 | 0 | |
| | | *Garra rufa* | Sea of Galilee | 8 | 8 | 3 | HUJ21009 |
| | | | Beit She'an streams | 4 | 3 | 0 | |
| | | | Ein T'eo | 6 | 3 | 0 | |
| | | *Garra nana* | Sea of Galilee | 2 | 2 | 0 | |
| | | *Carasobarbus canis* | Sea of Galilee | 9 | 6 | 0 | |
| | | | Ein T'eo | 5 | 5 | 0 | |
| | | *Hypophthalmichthys molitrix** | Sea of Galilee | 23 | 23 | 0 | |
| | | *Hypophthalmichthys molitrix* | Sea of Galilee | 6 | 6 | 0 | |
| | | *Pseudophoxinus kervillei* | Sea of Galilee | 1 | 1 | 0 | |
| | Nemacheilidae | *Oxynoemacheilus leontinae* | Sea of Galilee | 5 | 3 | 0 | |
| Cyprinodontiformes | Poeciliidae | *Gambusia affinis* | Sea of Galilee | 7 | 3 | 0 | |
| | Cyprinodontidae | *Aphanius mento* | Sea of Galilee | 1 | 1 | 0 | |
| Perciformes | Blenniidae | *Salaria fluviatilis* | Sea of Galilee | 10 | 8 | 0 | |
| Mugiliformes | Mugilidae | *Mugil cephalus* | Sea of Galilee | 10 | 7 | 0 | |
| | | *Chelon ramada* | Sea of Galilee | 13 | 9 | 0 | |
| Cichliformes | Cichlidae | *Astatotilapia flaviijosephi* | Beit She'an streams | 10 | 9 | 3 | HUJ21007 |
| | | *Coptodon zillii* | Sea of Galilee | 29 | 8 | 2 | HUJ21010 |
| | | *Tristramella simonis* | Sea of Galilee | 14 | 6 | 0 | |
| | | *Sarotherodon galilaeus* | Sea of Galilee | 120 | 17 | 9 | HUJ21012 |
| | | | Ein Afek | 19 | 4 | 0 | |
| | | *Oreochromis aureus* | Sea of Galilee | 11 | 7 | 2 | HUJ21011 |
| | | *Oreochromis aureus*** | Sea of Galilee | 1 | 1 | 0 | |

+ Not known to be found in the Sea of Galilee, closely related to *Acanthobrama lissneri*.

* Suspected hybrid of *Hypophthalmichthys molitrix* and *Hypophthalmichthys nobilis*.

** Suspected hybrid of *Oreochromis niloticus* and *Oreochromis aureus*.

for 2 min; further 23 cycles at the lower annealing temperature and final elongation step at 72˚C for 10 min. PCR products were verified by electrophoresis using 1.5% TBE Agarose gel containing ethidium bromide. PCR products were visualized using Gel Doc XR+ (BIO-RAD) and using ImageLab software (BIO-RAD). Successful PCR products were purified using an Exonuclease I-Shrimp Alkaline Phosphatase (ExoSAP-IT) method (USB, Cleveland, OH) and Sanger sequenced using BigDye V3.1 (Applied Biosystems) terminator chemistry in the forward and reverse directions on an ABI PRISM 3730xl DNA Analyzer (Applied Biosystems) following the manufacturers protocol.

## Data analysis

COI sequence files were assembled and aligned using clustalW algorithm with BioEdit software (Version 7.2.5) [38]. All the polymorphisms detected by the software were manually inspected on the chromatograms and dubious cases were manually removed. Successful sequences for each species together with their collection data were deposited into Barcode Of Life Data system (BOLD, https://www.boldsystems.org) [36] under project name LKCOX. See number of deposited sequences for the different species in Table 1. All sequences were aligned, manually inspected, and polymorphic sequences from the same species were defined as different haplotypes. Each haplotype for each species was queried against species level barcode records using BOLD identification module to derive molecular species identification. In some cases, the molecular identification did not completely match the original morphological species identification. For species where morphological and barcoding identification matched, only the barcodes of sampled fish were considered. For species with disagreements, also barcodes of the same and related species were downloaded from BOLD and considered in further sequence-based analyses. From the aligned sequences, an estimated unrooted gene tree, based on 530 bp long COI haplotypes, was constructed utilizing the Molecular Evolutionary Genetics Analysis (MEGA6) freeware [39], using the Kimura-2-parameter model, and the neighbor-joining tree construction method with 1000 bootstrap replicates [40]. The tree with the strongest bootstrap support was chosen for representation. Based on all COI haplotypes and using MEGA6 software, K2P genetic distance between any pair of haplotypes was calculated. The average pairwise differences were manually calculated by summing up and averaging the pairwise haplotype distances within and between different taxa. Fish that were suspected to be hybrids according to their morphology were excluded from the average distance calculations.

## Ethical approval

Fish sampling from protected nature reserves was done under special permits 2012/38733 and 2014/40233 for sampling protected wildlife as reviewed and approved by the Israel Nature and National Parks Protection Authority. Some samples were obtained from catches of a commercial fishing vessel.

## Results

### Molecular barcoding

Samples and barcode sequences were obtained for a total of 22 fish species, including16 out of the 18 native and five non-native species, reported in the Sea of Galilee [16, 19, 20]. In addition, *Acanthobrama telavivensis* Goren, Fishelson & Trewavas, 1973, a species not reported in the lake, was sampled outside the Sea of Galilee for comparison to the closely related *Acanthobrama lissneri* Tortonese, 1952 [41], which is one of the native fish species that was not captured in any of the Sea of Galilee samplings. Another native fish species absent from this study, *Tristramella sacra* (Gunther, 1865), is considered to be extinct [42]. Missing from the non-native species were *Anguilla* and *Xiphophorus hellerii* that were not captured in any of the Sea of Galilee samplings. Altogether, COI sequences were analyzed from 176 individual fish, belonging to 22 freshwater species (Table 2). Additionally, seven sequences were analyzed from two fish species, *H. molitrix* (six sequences) and *Oreochromis aureus* (Steindachner, 1864) (one sequence), that were suspected as hybrids with aquaculture fish (hereafter referred to as 'suspected hybrids') (Table 2).

For 20 out of 22 species more than one specimen was collected and sequenced, and out of those, 12 species had more than one barcode sequence variant (COI haplotypes). The K2P

**Table 2. Summary of fish species, number of barcoded samples, COI haplotypes naming, haplotype frequencies and within species genetic distances.**

| Order | Family | Species | N successfully barcoded samples | Haplotype | Haplotype frequency | K2P distance within species |
|---|---|---|---|---|---|---|
| Siluriformes | Clariidae | *Clarias gariepinus* | 5 | CgA | 1 | 0 |
| Cypriniformes | Cyprinidae | *Mirogrex terraesanctae* | 8 | AtA | 0.625 | 0.0019 |
| | | | | AtB* | 0.125 | |
| | | | | AtC | 0.125 | |
| | | | | AtD | 0.125 | |
| | | *Luciobarbus longiceps* | 7 | BlA | 0.571 | 0.0019 |
| | | | | BlB | 0.429 | |
| | | *Acanthobrama telavivensis+* | 7 | AtlA | 0.571 | 0.0019 |
| | | | | AtlB* | 0.429 | |
| | | *Capoeta damascina* | 5 | CdA | 0.4 | 0.0038 |
| | | | | CdB | 0.4 | |
| | | | | CdC* | 0.2 | |
| | | *Cyprinus carpio* | 5 | CcA | 1 | 0 |
| | | *Garra rufa* | 14 | GrA | 0.571 | 0.0086 |
| | | | | GrB* | 0.214 | |
| | | | | GrC | 0.143 | |
| | | | | GrD* | 0.072 | |
| | | *Garra nana* | 2 | GnA | 1 | 0 |
| | | *Carasobarbus canis* | 11 | BcA | 0.909 | 0.0019 |
| | | | | BcB* | 0.091 | |
| | | *Hypophthalmichthys molitrix* | 23 | HmA | 0.783 | 0.0057 |
| | | | | HmB | 0.217 | |
| | | *Hypophthalmichthys molitrix*** | 6 | HmA | 0.333 | 0.0362 |
| | | | | HmB | 0.5 | |
| | | | | HmC | 0.167 | |
| | | *Pseudophoxinus kervillei* | 1 | PkA* | 1 | 0 |
| | Nemacheilidae | *Oxynoemacheilus leontinae* | 3 | NlA* | 0.667 | 0.0019 |
| | | | | NlB* | 0.333 | |
| Cyprinodontiformes | Poeciliidae | *Gambusia affinis* | 3 | GaA | 1 | 0 |
| | Cyprinodontidae | *Aphanius mento* | 1 | AmA* | 1 | 0 |
| Perciformes | Blenniidae | *Salaria fluviatilis* | 8 | SfA | 0.875 | 0.0038 |
| | | | | SfB* | 0.125 | |
| Mugiliformes | Mugilidae | *Mugil cephalus* | 7 | McA | 1 | 0 |
| | | *Chelon ramada* | 9 | LrA | 0.889 | 0.0019 |
| | | | | LrB | 0.111 | |
| Cichliformes | Cichlidae | *Astatotilapia flaviijosephi* | 9 | AfA | 0.778 | 0.0025 |
| | | | | AfB* | 0.111 | |
| | | | | AfC | 0.111 | |
| | | *Coptodon zillii* | 8 | TzA | 1 | 0 |
| | | *Tristramella simonis* | 6 | TsA | 0.833 | 0.0019 |
| | | | | TsB | 0.167 | |
| | | *Sarotherodon galilaeus+* | 21 | SgA | 1 | 0 |
| | | *Oreochromis aureus* | 7 | OaA | 1 | 0 |

(*Continued*)

**Table 2.** (Continued)

| Order | Family | Species | N successfully barcoded samples | Haplotype | Haplotype frequency | K2P distance within species |
|---|---|---|---|---|---|---|
| | | *Oreochromis aureus*\*** | 1 | OaB | 1 | 0 |

Here the species of fish is based on morphological taxonomy. + Not known to be found in the Sea of Galilee, closely related to *Acanthobrama lissneri*.

\* Newly deposited haplotypes.

\** Suspected hybrid of *Hypophthalmichthys molitrix* and *Hypophthalmichthys nobilis*.

\*** Suspected hybrid of *Oreochromis niloticus* and *Oreochromis aureus*.

distances within species that were calculated for each species ranged between 0–0.0086 with a mean of 0.0017 (Table 2). For ten species with barcodes existing in BOLD, 12 new barcodes were added. These added haplotypes were always those with the lower frequency in our data (Table 2). For 19 out of 22 species, the molecular identification agreed with the original morphological identification, including for species with more than one COI haplotype.

## Disagreements with molecular barcoding

Three species had disagreements between morphological and molecular barcoding identification. Further investigation was carried out adding all available COI barcode records from BOLD of query and related species. For the species which was not reported in the Sea of Galilee, *Acanthobrama telavivensis*, fish sampled from Ein Afek spring pools had two very similar haplotypes (99.8% sequence identity), with similar frequencies, one of which (AtlB) was new to BOLD. Five records were available from BOLD, four listed for *A. lissneri* and one for *A. telavivensis*, but all sharing an identical sequence. Thus, *A. telavivensis* fish with the haplotype common to our data and BOLD (AtlA), were identified by BOLD with 100% fit for both *A. lissneri* (a species native to the Sea of Galilee) and *A. telavivensis* (not reported in the Sea of Galilee).

Three *Gambusia affinis* collected samples, all with the same haplotype (GaA), were identified by BOLD as *Gambusia holbrooki* Girard, 1859 (100% fit), followed by *G. affinis* (99.81% fit). In the analysis of all available *G. holbrooki* and *G. affinis* records (BOLD and GaA), the GaA haplotype of sampled fish clustered together with *G. holbrooki* records of fish from widely distributed places.

*Garra rufa* was represented by 14 samples in our collection, showing four haplotypes, of which two were new to BOLD. BOLD identified *G. rufa* samples of this study as *Garra jordanica* Hamidan, Geiger & Freyhof, 2014, with 98.7–100% sequence identity for different haplotypes. This disagreement will be analyzed further later under its own subsection.

Molecular identification using BOLD for samples of fish suspected as hybrids provided genetic evidence to support hybridization. A sample suspected as *Oreochromis niloticus* (Linnaeus, 1758) x *Oreochromis aureus* hybrid, which is neither native nor a hybrid stocked into the Sea of Galilee, had a COI sequence with a perfect match (100%) to *O. niloticus*, indicating that this fish is at least partly hybridized with *O. niloticus*, a non-native species not reported to inhabit the Sea of Galilee. Additionally, in samples from the Sea of Galilee that according to their morphology were suspected to be *Hypophthalmichthys molitrix* hybridized with *Hypophthalmichthys nobilis* (Richardson, 1845), three different COI haplotypes were found. Two of them matching BOLD *H. molitrix*, reported to be stocked into the lake, while the third matching BOLD *H. nobilis* (100%), not reported in the lake, supporting possible hybridizations in both directions. K2P distance within these hybrid fish was 0.0362, over 4-fold higher than the maximum distance found within the other species, supporting that these fish were hybrids.

## Genetic distances and phylogeny from barcoding data

To evaluate the DNA barcoding efficiency in distinguishing among species, K2P genetic distances between each pair of species was calculated. The genetic distance between species ranged between 0.0095–0.2886, with a mean distance of 0.2295, over 100-fold higher compared to the mean within species K2P distance (0.0017). Additionally, the range of between-species distances did not overlap with that of within-species, supporting the utility of COI haplotypes in differentiation of these species. The species differentiation is evident also from the phylogenetic tree based on the COI haplotypes (Fig 1). In the tree, species were grouped similarly to the expected tree based on the NCBI taxonomy browser (https://www.ncbi.nlm.nih.gov/Taxonomy/taxonomyhome.html/). The phylogeny separated well all the different species, while grouping together multiple haplotypes of the same species. Furthermore, on the higher taxonomic levels, the phylogeny separated well the three most represented families in the Sea of Galilee, cyprinids, cichlids and mugilids.

However, a few discrepancies were found also in the genetic distances and phylogeny data. A genetic distance of 0.08 was found between barcodes of *Acanthobrama lisnneri/Acanthobrama telavivensis* and *Mirogrex terraesanctae*, similar to the within-genus distance found between *Garra rufa* and *Garra nana* (Heckel, 1843).

Additionally, within the *Cichlidae* clade in the neighbor-joining phylogenetic tree (Fig 1), *Oreochromis aureus* (Haplotype OaA) clustered together with species from a different genus, *S. galilaeus* (haplotype SgA), rather than with the species with which its genus is shared, *O. niloticus* (haplotype OaB). Support for this genetic similarity comes from a smaller K2P distance between *O. aureus* and *S. galilaeus* (0.0095) compared to distances between genera of other Sea of Galilee cichlids (0.06–0.16), and genera of cyprinids (0.08–0.23) (Fig 1).

## Species in the *Garra* genus

Comparison of sampled *Garra rufa* fish haplotypes to BOLD database accessions gave inconsistencies in species definition. Also, K2P distance within haplotypes of fish from Israel, identified as *G. rufa* in this study, was 0.0086, the highest value found within species groups and more than four times higher than the mean within-species value (0.0014) of all other species groups excluding *G. rufa* haplotypes. *Garra rufa* fish were collected in this study from three places: Sea of Galilee, Asi stream (one of Beit She'an streams) and Ein Te'o (North of the Sea of Galilee) (Fig 2A). These fish had four COI haplotypes (GrA–GrD) that were unevenly distributed between collection sites. Among Sea of Galilee samples, five, two and one fish had haplotypes 'GrA', 'GrC' and 'GrD', respectively. All three fish from Ein Te'o had haplotype 'GrA' and all three fish from Asi stream had haplotype 'GrB' (Fig 2A).

Sequences of 16 *Garra* species samples from this study together with 33 BOLD accessions, which had geographic location information assigned to them (see S1 Table), were aligned to construct a neighbor-joining phylogenetic tree (Fig 2B). Haplotypes of *G. nana* were separated from all other haplotypes defined as *G. rufa*, *G. jordanica* and *Garra ghorensis* Krupp, 1982. The two haplotypes of *G. nana* (GnA and GnB) were relatively distinct from one another (K2P = 0.0381; Table 3). GnA, was found in fish from the Sea of Galilee and the more northern location in Syria, while GnB, was found in fish from an isolated location more southern in Syria (Fig 2A).

Israel *G. rufa* samples from the Sea of Galilee and Ein Te'o spring had haplotypes GrA, GrC and GrD. These haplotypes of *G. rufa* fish from Israel clustered with BOLD haplotype Gj of *G. jordanica* fish sampled from streams in the northern basin of the Dead Sea in Jordan (Fig 2B). Close by in the phylogeny, yet separated, was haplotype GrB of *G. rufa* fish from Asi stream, again similar to *G. jordanica* haplotype, but somewhat genetically separated from haplotypes

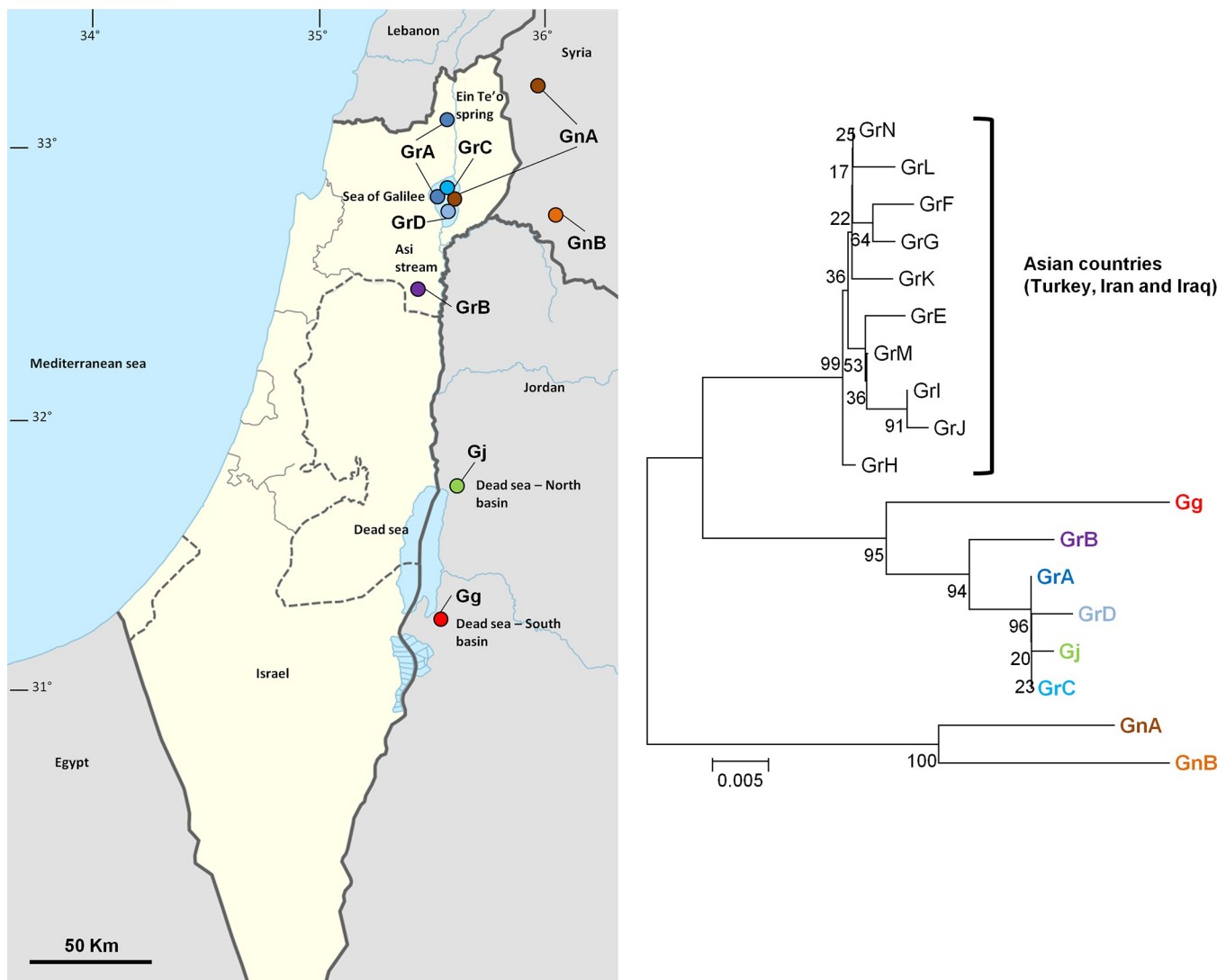

**Fig 2. Sampling sites and clustering of *Garra* samples based on COI haplotypes.** Acronyms of populations are: GrA–*Garra rufa* from haplotype A; GrB–*Garra rufa* from haplotype B; GrC–*Garra rufa* from haplotype C; GrD–*Garra rufa* from haplotype D; Gg–*Garra ghorensis* from the southern basin of the Dead sea; Gj–*Garra Jordanica* from the northern basin of the Dead sea; GnA–*Garra nana* from haplotype A; GnB–*Garra nana* from haplotype B. (**A**) Map showing the sampling sites in a regional context. Different *Garra* haplotypes were denoted by differently colored circles in each sampling site on the regional map. Map was reprinted from DMY + NordNordWest (https://commons.wikimedia.org/wiki/File:Israel_by_israeli_law_adm_location_map.svg), under a CC BY-SA 4.0 license, original copyright 2017. (**B**) A neighbor-joining unrooted tree constructed for haplotypes obtained from samples collected in this study alongside haplotypes mined from BOLD. Next to the edges are haplotype acronyms as indicated above and next to bifurcations are percent bootstrap support values. Haplotype text colors in the tree match circle colors in the map (A).

of the Sea of Galilee population. Further separated from this cluster of haplotypes (Israeli *G. rufa* and non-Israeli *G. jordanica*) was the BOLD *G. ghorensis* haplotype (Gg) of fish from streams in the southern basin of the Dead Sea. Further away, clustered together were BOLD haplotypes (GrE–GrN) of fish defined as *G. rufa* from other Asian countries (GrAsia—Turkey, Iran and Iraq).

Given the genetic divergence found for geographically isolated fish, pairwise K2P distances were calculated for *Garra* and compared to the distances found within and between species based on all species (Table 2, Fig 1). For each *Garra* combination of species and geographic location the mean, minimum and maximum values of the pairwise K2P distances were

**Table 3. Genetic distances between geographically separate *Garra* populations.**

| | Gr North | Gr Asi | Gj | Gg | Gr Asia | GnA | GnB |
|---|---|---|---|---|---|---|---|
| Gr North | 0.0025 | | | | | | |
| | 0–0.0037 | | | | | | |
| Gr Asi | 0.0143 | 0 | | | | | |
| | 0.013–0.0168 | | | | | | |
| Gj | 0.0031 | 0.0149 | 0 | | | | |
| | 0.0018–0.0055 | | | | | | |
| Gg | 0.0414 | 0.0381 | 0.0421 | 0 | | | |
| | 0.0401–0.0441 | | | | | | |
| Gr Asia | 0.0462 | 0.0469 | 0.0469 | 0.0587 | 0.0065 | | |
| | 0.0422–0.0523 | 0.0442–0.0502 | 0.0442–0.0502 | 0.0563–0.0604 | 0.0018–0.013 | | |
| GnA | 0.0792 | 0.0857 | 0.0814 | 0.0834 | 0.0635 | 0 | |
| | 0.0792–0.0792 | | | | 0.0582–0.0685 | | |
| GnB | 0.0899 | 0.0965 | 0.0921 | 0.0854 | 0.0692 | 0.0381 | 0 |
| | 0.0899–0.0899 | | | | 0.0664–0.727 | | |

In each table cell, top row is the mean K2P value and bottom row is the range of pairwise distances. Acronyms for populations are: GrNorth–*Garra rufa* from Sea of Galilee and Ein Te'o of Northern Jordan valley; GrAsi–*Garra rufa* from Asi atream of Beit She'an valley; Gj–*Garra jordanica* from the northern basin of the Dead Sea; Gg–*Garra ghorensis* from the southern basin of the Dead Sea; GrAsia–*Garra rufa* from Asian countries including Turkey, Iran and Iraq; GnA–*Garra nana* from Haplotype A; GnB–*Garra nana* from Haplotype B.

calculated (Table 3). It is evident that the genetic separation between populations of *G. rufa* increases with geographic distance. The Israeli *G. rufa* fish, are genetically divergent from *G. rufa* fish from farther eastern and northern Asian countries (Mean K2P between Gr North and Gr Asia = 0.0462, and between Gr Asi and Gr Asia = 0.0469; Table 3). The mean K2P value between these *G. rufa* clades is considerably higher than what was observed as within-species differences (Table 2), and more similar to between-species ones (Fig 1). On the other hand, mean K2P distance between *G. jordanica* BOLD haplotype and Israeli *G. rufa* fish (Gr North; Fig 2) was very low (0.0031, Table 3) and well within the range of within-species distances (Table 2).

Among *G. rufa* fish from Israel, Asi stream fish of Beit she'an valley (Gr Asi) had a slightly different haplotype (GrB) than neighboring fish from Sea of Galilee and Ein T'eo spring (mean K2P between Gr Asi and Gr North = 0.0143) and thus, were also slightly genetically different from BOLD samples defined as *G. jordanica* (mean K2P between Gr Asi and Gj = 0.0149) (Table 3). These mean K2P distances are slightly higher than within-species distances (Table 2) and considerably lower than between-species differences (Fig 1).

Samples from BOLD of *G. ghorensis* fish from the southern part of the Jordan valley were similarly genetically distant from Asian samples of *G. rufa* (mean K2P between Gg and Gr Asia = 0.0587) and from Israeli *G. rufa* samples (mean K2P between Gg and Gr North = 0.0414 and between Gg and Gr Asi = 0.0381) (Table 3). Finally, samples of *G. nana* were genetically distant from *G. rufa*, *G. jordanica* and *G. ghorensis* (Table 3), at levels similar to between-species distances (Fig 1). From all *Garra* populations, *G. nana* samples were the least distant from Asian *G. rufa* samples (Table 3).

## Discussion

The development of molecular methods and their availability have been driving an effort to augment morphological taxonomy of species with molecular identification based on DNA

sequencing [29, 43, 44]. Since within fish there are species-rich and diverse taxonomic groups, DNA barcoding has great potential to improve the taxonomy within these groups [30, 45]. In a previous study conducted by [46], the cichlids of Israel were barcoded with success. In this study, focus was given to the fish of the Sea of Galilee and surrounding water bodies in the region, including species other than cichlids. Altogether, COI sequences from 177 individual fish, belonging to 22 morphologically-defined freshwater species, were analyzed. From this study, 12 new barcode haplotypes were added to the public database, belonging to 10 of the 16 native species captured. Thus, minor frequency new barcodes were identified for a high proportion of the native species, an indication for the genetic divergence and uniqueness of these regional native populations. For 19 out of the 22 species, morphological and DNA barcoding agreed on the species definition, except for a few discrepancies which will be discussed later on. In theory, the 22 species, as they were defined by morphological taxonomy, should have returned 22 DNA barcodes, but in practice, 40 different COI haplotypes were recovered. For 12 species, mostly belonging to the Cyprinidae and Cichlidae, two or more COI haplotypes were found, indicating that genetic variation had evolved not exactly at the same rate as species did. Once more populations of these species from different locations will be studied, a comparison to this study could highlight if this intraspecific variation represents the beginning of population/species divergence or is this variation ancestral/common to the species.

The mean K2P genetic distance within species was small, corroborating within-species ranges in other studies [26, 29, 45, 47]. Furthermore, the range of genetic distances between species did not overlap the range within species. Therefore, importantly, for the Sea of Galilee, COI sequencing provides a reliable method to identify species and study their ecology.

Defining the expected K2P genetic distance (mean and range) for different haplotypes within and between species (or higher taxonomic levels), can provide a useful tool for ecological and taxonomic research [29, 30]. Here we used phylogeny, genetic distances and geographic information to study a few discrepancies identified between morphological taxonomy and DNA barcoding-based taxonomy. Two species of the same genus, *A. lissneri*, native to the Sea of galilee, and *A. telavivensis*, reported only outside the lake, shared one of the barcodes found for *A. telavivensis*. The other barcode of *A. telavivensis* was also very similar, suggesting high genetic similarity of all available barcodes and thus, possible reconsideration of these fish taxonomy. Additionally, a relatively small genetic distance, typical of distances between species, was found between *A. lissneri/telavivensis* and *M. terraesanctae*, two species of different genera, suggesting these genera are closer than expected. Thus, smaller than expected molecular divergence was found both between species of the same genera and between species of different genera, all belonging to this family. It should be noted that the genus of *Mirogrex terraesanctae* was originally defined as *Acanthobrama terraesanctae* [41, 48] and the molecular data supports the original classification. Another species, the mosquito fish *G. affinis*, invasive to the Sea of Galilee, had an identical barcode to BOLD barcodes of *G. holbrooki* from various other locations. As these fish were introduced into several freshwater bodies to combat mosquito-borne diseases, it might be worthwhile to reconsider if indeed they deserve to be separate species and if they do, then the fish in the Sea of Galilee might be *G. holbrooki* rather than *G. affinis*.

Morphological taxonomy of African cichlids, a very species-rich group of fish, was occasionally revisited [49–51]. Cichlids in the Sea of Galilee are of African origin and they constitute a key group in the lake, with both ecological and commercial significance [16]. Here, phylogeny proximity and a small genetic distance, on the order of between species of the same genera, were found between *O. aureus* and *S. galilaeus*, two species from different genera. This level of molecular relatedness was considerably smaller compared to the distance between *O. aureus* and *O. niloticus*, which share the same genus. Thus, such and other molecular methods can help further revisiting and updating cichlid phylogeny [50]. Furthermore, in this study a fish morphologically

suspected as a hybrid between *O. niloticus* and *O. aureus* was molecularly identified as *O. niloticus*, supporting hybrids might exist in the lake [22], since pure *O. niloticus* are considered too cold-temperature sensitive to become established there [52]. Also, indication for the presence of hybrid fish between *H. molitrix* and *H. nobilis* were found. Both these specific cichlid and carp hybrids are commonly used in Israeli aquaculture and as such, fish might have somehow reached into the Sea of Galilee by unintentional introduction or as escapees from nearby ponds.

Phylogeny, genetic distances and geographic information were mainly employed for studying the discrepancies found in regional *Garra* taxonomy, for which COI sequences of samples collected in this study were analyzed together with sequences from the public BOLD database. Presence of *G. rufa* and *G. nana* has been reported in the upper Jordan valley system (including in the Sea of Galilee, Hula valley and Beit She'an valley streams) [16, 19]. *Garra jordanica* and *Garra ghorensis* were reported as endemic to freshwater springs and streams northern and southern to the Dead Sea, respectively [53, 54]. The phylogenetic tree based on COI sequences suggested some level of geographic separation between groups of haplotypes.

The genetic divergence found between *G. rufa* from the Asi stream (Beit She'an valley) south to the Sea of Galilee and the more northern Israeli fish (those from the Sea of Galilee itself and from the Hula valley) was the highest of within-species distances. This level of genetic divergence within the northern Israeli fish, is likely due to the recent (~24,000 years ago) isolation of the Beit She'an valley water system from the Jordan River system [55].

Furthermore, *G. rufa* from Turkey, Iran and Iraq formed a cluster separated from Mediterranean *Garra* samples. The range of K2P distances (0.0422–0.0523) suggests that Asian *G. rufa* and Mediterranean *G. rufa* should be considered as different species. A recent study suggested that fish described as *G. rufa* from the springs and streams northern to the Dead Sea in Jordan, are in fact a separate species described as *G. jordanica* [53]. As found here, Israeli fish defined as *G. rufa* are genetically more similar to BOLD *G. jordanica* than to *G. rufa* from Asian countries. K2P distances between Israeli *G. rufa* haplotypes and BOLD *G. jordanica* haplotype ranged between 0.0018–0.0149, falling well inside the within-species range. Additionally, in the phylogentic tree, *G. jordanica* haplotype clustered together with haplotypes of *G. rufa* from Israel, consistent also with their geographic proximity. Thus, given the distance from Asian *G. rufa* and proximity to *G. jordanica*, a change in species definition for *G. rufa* fish from Israel, from *G. rufa* to *G. jordanica*, should be considered. This was also suggested theoretically by [53] based on the geographic distribution data, and is now supported by DNA barcoding evidence. The analyses of discrepancies between morphological and molecular species definition, primarily that of the *Garra* complex, are examples for how DNA barcoding can assist in resolving and revisiting taxonomic and ecological questions.

The utility of species identification using DNA barcoding in the context of ecological research was extensively discussed in the last decade and new implementations of DNA barcoding are being published ever since this method was standardized [28, 44, 56, 57]. The sampling scheme employed for this study allowed for the first time to develop and apply molecular genetics tools to many of the freshwater fish species of Israel in the context of their natural habitats. Studies are starting to employ molecular tools relying on or assisted by molecular species identification to address ecological questions regarding native fish populations in the Sea of Galilee, such as the spatial and temporal distribution and abundance of cichlid fry along lake shores [58] and the decline of genetic variation in the *S. galilaeus* lake population [27].

## Conclusions

This study established a molecular DNA barcoding database for the ecologically isolated native and non-native fish species inhabiting the Sea of Galilee. The data obtained here exemplified

how in combination with the public BOLD database and analyses of geographic patterns, morphological and molecular identification can address issues in species definition on one hand, and on the other, how the molecular data could be used as a basis to address ecological questions concerning freshwater fish assemblages. Developing modern, molecular tools for identification of fish down to species level and even further, to population level, are being valuable tools for monitoring fish populations and supporting management and conservation decisions of significant populations in key freshwater bodies as the Sea of Galilee.

## Supporting information

**S1 Table. *Garra* species accessions used for analyses.** Garra species accessions and geographic information for 16 sample sequences from this study and 33 BOLD accessions. (XLSX)

## Acknowledgments

The authors wish to thank all who assisted in sampling of fish: David Cummings (Kinneret Limnological Laboratory, IOLR), Dr. Noam Leader, Dr. Dana Milstein, Dr. Amit Dolev and Gisele Hazzan (Israel National Parks Authority), Menachem Lev (Kibbutz Ein Gev fishing vessel) and colleagues from the laboratory and the Animal Sciences Department. LD is chair of the Vigevani Senior Lectureship in Animal Sciences.

In memory of our dear friend and colleague Tomer Borovski.

People always tend to shine in our memories after they have passed, however, Tomer was shining just as bright when he was alive–an extraordinary person, a friend of people with a big heart and a great sense of humor. A biologist and a multidisciplinary autodidact with vast knowledge that he was always happy to share.

## Author Contributions

**Conceptualization:** Tomer Borovski, Lior David.

**Data curation:** Roni Tadmor-Levi, Tomer Borovski, James Shapiro, Daniel Golani, Lior David.

**Formal analysis:** Roni Tadmor-Levi, Tomer Borovski, Evgeniya Marcos-Hadad, James Shapiro, Lior David.

**Funding acquisition:** Gideon Hulata, Lior David.

**Investigation:** Roni Tadmor-Levi, Tomer Borovski.

**Methodology:** Roni Tadmor-Levi, Evgeniya Marcos-Hadad, Lior David.

**Project administration:** Gideon Hulata, Lior David.

**Resources:** Tomer Borovski, James Shapiro, Daniel Golani, Lior David.

**Supervision:** Gideon Hulata, Lior David.

**Writing – original draft:** Roni Tadmor-Levi, Tomer Borovski, Lior David.

**Writing – review & editing:** Roni Tadmor-Levi, Daniel Golani, Lior David.

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
