## [Decision Letter · Decision Letter 0]

9 Nov 2021

PONE-D-20-39138Establishing and using a genetic database for resolving identification of fish species in the Sea of Galilee, IsraelPLOS ONE

Dear Dr. David,

Thank you for submitting your manuscript to PLOS ONE. After careful consideration, we feel that it has merit but does not fully meet PLOS ONE’s publication criteria as it currently stands. Therefore, we invite you to submit a revised version of the manuscript that addresses the points raised during the review process.

My apologies that this review has taken so long. I was assigned this manuscript after the first academic editor stepped down. We had a difficult time finding reviewers during the COVID pandemic. Please revise as per the second reviewer.

We look forward to receiving your revised manuscript.

Kind regards,

Mary Peacock, Ph.D.

Academic Editor

PLOS ONE

Journal Requirements:

2. Our internal editors have looked over your manuscript and determined that it is within the scope of our Freshwater Ecosystems Call for Papers. This collection of papers is headed by a team of Guest Editors for PLOS ONE (https://collections.plos.org/s/freshwater-ecosystems). The Collection will encompass a diverse range of research articles on freshwater ecology, including freshwater fish ecology. Additional information can be found on our announcement page: https://collections.plos.org/s/freshwater-ecosystems.

If you would like your manuscript to be considered for this collection, please let us know in your cover letter and we will ensure that your paper is treated as if you were responding to this call. If you would prefer to remove your manuscript from collection consideration, please specify this in the cover letter.

4. We note that Figure 2 in your submission contain map images which may be copyrighted. All PLOS content is published under the Creative Commons Attribution License (CC BY 4.0), which means that the manuscript, images, and Supporting Information files will be freely available online, and any third party is permitted to access, download, copy, distribute, and use these materials in any way, even commercially, with proper attribution. For these reasons, we cannot publish previously copyrighted maps or satellite images created using proprietary data, such as Google software (Google Maps, Street View, and Earth). For more information, see our copyright guidelines: http://journals.plos.org/plosone/s/licenses-and-copyright.

Reviewers' comments:

Reviewer's Responses to Questions

**Comments to the Author**

1. Is the manuscript technically sound, and do the data support the conclusions?

Reviewer #1: Yes

Reviewer #2: Yes

2. Has the statistical analysis been performed appropriately and rigorously? 

Reviewer #1: Yes

Reviewer #2: Yes

3. Have the authors made all data underlying the findings in their manuscript fully available?

Reviewer #1: Yes

Reviewer #2: Yes

4. Is the manuscript presented in an intelligible fashion and written in standard English?

Reviewer #1: Yes

Reviewer #2: No

5. Review Comments to the Author

Reviewer #1: This paper describes very well the DNA barcoding of the freshwater fish found in Israel (mainly the Sea of Galileea). All required analyses were well done and I only found two digitation mistakes in species names: line 304 ghorensis, and line 353: Garra (italic). I also do not think that the authors should use names as Garra spp. because this ssp. means species of this genus and when we cite the genus we are already talking about its species.

Reviewer #2: **Please refer to attachment for the full review which includes major and minor corrections.** Below are general comments and suggestions for each section of the manuscript.

Abstract

The authors summarize the main research questions and key findings very well. Using the abstract as a foundation, the introduction and discussion should highlight these points in order to strengthen them.

Introduction

DNA barcoding was used to establish a database of molecular species identification based on sequences of CO1 gene for the Sea of Galilee, Israel. Applied DNA barcoding to samples taken mainly from the Sea of Galilee to address taxonomic questions and build a reference barcoding database for molecular identification of fish in the context of the Sea of Galilee and the region. The study should more explicitly indicate that it is not building a new database but rather adding to an existing database such as BOLD. More information of the BOLD database should be included in the introduction as well as the standardised rules that apply to DNA barcoding. The study fails to relate to previous research in this area. The authors should include more examples, information and scenarios as well as reference the related literature, especially by including more recently published work. Hence, a stronger link between the literature and the topic of this manuscript is required. In the materials and methods section, you mention that methods were adopted from Borovski et al. (2018), however, no mention of this manuscript was included in the Introduction and Lines 71-74 were directly taken from it with no changes. The main point of the study which is to identify fish species from the Sea of Galilee and surrounding regions using DNA barcoding to build a reference database for effective management/conservation does not come across strongly. In other words, the exact reason for this study other than building a database is not explicitly clear, is it for the management of invasive species, monitoring of native species? The link to this being mentioned earlier in the introduction needs to be made clearer towards the end of the introduction.

Methods

The experiments or intervention used was DNA barcoding. DNA barcoding was appropriate for addressing the research question. However, the data collected was not interpreted accurately. I found it very hard to follow the protocols and feel that another researcher would battle to reproduce this study based on the methodology reported. For instance, sampling methods are not clear, number of samples collected for each species per site are not clear as well as how many successful sequences were used from this sample size. No mention of whether the fish collected were stored in formalin for taxonomic identification as well as no mention of photos of samples for each species were taken to upload with BOLD. References were not provided or corresponded to the authors point. Data analysis section needs work. Please could the authors rectify the above and include them into the materials and methods section

Results

Results are very interesting! Figures and tables are clear and readable, however Figure 2 is not of great quality and more information needs to be incorporated in to the map (Fig 2A; see below corrections for further recommendations). Table 2 legend should be above the table and not below it. I would recommend for Figure 1 to exclude the K2P distances combined here and provide them as separate tables as has been done for Table 2, just separate it according to family using sub-headings. Also correct acronyms for Figure 2 and Table 2 (see below corrections for further recommendations).

Discussion

The discussion section is almost a repetition of the authors results section. More evidence needs to be provided here to support your results. Elaborate by including more examples of relevant recent literature to strengthen your arguments. Results are interesting, however a lot more work needs to be done in this section. Also, it is important to include any limitations that were raised during this this study and should be highlighted and discussed here by providing ways in the future as to how it can be avoided or improved on. Also, there is no conclusion section, which is very important to drive your take home message for this study. For instance, in the conclusion section by adding a how this study could contribute to the conservation of Israel’s native species and identification of hybrid and non-native invasive species through DNA barcoding would be a good as well as how it will improve in the future from this perspective.

Furthermore, before discussing the G. rufa complex in the discussion, authors need to discuss the other disagreements that were found from your results, namely A. lissneri and A. telavivensis; G. holbrooki and G. affinis; hybrids; Cichlidae; and Acanthobrama-Mirogrex, in more detail by referring to or by providing recent, relevant and supportive literature. In the results section, I indicated which sentences need to be moved to the discussion section relative to the above sentence as they should not be present in the results section.

6. PLOS authors have the option to publish the peer review history of their article (what does this mean?). If published, this will include your full peer review and any attached files.

Reviewer #1: **Yes: **Claudio de Oliveira

Reviewer #2: No

---

## [Author Response · Author response to Decision Letter 0]

16 Jan 2022

The authors would like to thank the editor and reviewers for their constructive detailed comments. We followed these comments one by one revising the text accordingly. It is evident from this letter of response and also from the very busy 'track changes' version. Following these useful suggestions, we now feel that the revised version is a significant improvement over the initial submission to the level ready for publication. Bellow is a point-by-point response to the specific comments (Also provided as a separate file). 

Response to editor’s points:

1. We went over the manuscript and accompanied files to ensure they meet PLOS ONE style requirements. The revised files are submitted.

2. We would like our manuscript to be considered for the Freshwater Ecosystems collection.

3. We publicly released the repository data as stated in the Data Availability section.

4. Figure 2 was edited and provided at higher resolution. Additionally, we added a copyright statement in figure 2 legend.

Reviewer #1

Comment: This paper describes very well the DNA barcoding of the freshwater fish found in Israel (mainly the Sea of Galileea). All required analyses were well done and I only found two digitation mistakes in species names: line 304 ghorensis, and line 353: Garra (italic). I also do not think that the authors should use names as Garra spp. because this ssp. means species of this genus and when we cite the genus we are already talking about its species.

Response: We appreciate the recommendation. The two digitations mistakes were fixed and spp. was deleted from the text.

Reviewer #2: 

Abstract

The authors summarize the main research questions and key findings very well. Using the abstract as a foundation, the introduction and discussion should highlight these points in order to strengthen them.

Response: Accordingly, we almost did not change the abstract, except for a few sentences to emphasizing the points recommended by the reviewer in the discussion section.

Introduction

DNA barcoding was used to establish a database of molecular species identification based on sequences of CO1 gene for the Sea of Galilee, Israel. Applied DNA barcoding to samples taken mainly from the Sea of Galilee to address taxonomic questions and build a reference barcoding database for molecular identification of fish in the context of the Sea of Galilee and the region. 

Comment: The study should more explicitly indicate that it is not building a new database but rather adding to an existing database such as BOLD. More information of the BOLD database should be included in the introduction as well as the standardised rules that apply to DNA barcoding. The study fails to relate to previous research in this area. The authors should include more examples, information and scenarios as well as reference the related literature, especially by including more recently published work. Hence, a stronger link between the literature and the topic of this manuscript is required. In the materials and methods section, you mention that methods were adopted from Borovski et al. (2018), however, no mention of this manuscript was included in the Introduction and Lines 71-74 were directly taken from it with no changes. The main point of the study which is to identify fish species from the Sea of Galilee and surrounding regions using DNA barcoding to build a reference database for effective management/conservation does not come across strongly. In other words, the exact reason for this study other than building a database is not explicitly clear, is it for the management of invasive species, monitoring of native species? The link to this being mentioned earlier in the introduction needs to be made clearer towards the end of the introduction.

Response: see below a point-by-point response.

Corrections:

Comment: L50-L51: Remove sentence.

Response: Sentence was removed and replaced by the suggested one.

Comments: L55: Please add a reference for the last bit of this sentence and change the sentence to read as follows, “Despite freshwater habitats only comprising of 1% of the Earth’s surface water, they support 43% (13 000 species) of the world’s freshwater fish [2,5], rich in biodiversity across all trophic levels [reference, e.g. Vörösmarty et al., 2010; Carpenter et al., 2011]”.

Response: sentence was changed as suggested.

Comment: L55-L57: Remove this sentence, as the content has been incorporated in the previous sentence at the beginning of line 55. Also, according to Leveque et al. (2007), only freshwater species and peripheral species add up to 13 000 and 15 000 if brackish water fish are included. According to the authors for this review, “15 000 freshwater fish species are known around the world”. This is incorrect, please change the sentence to read either as “13 000 freshwater fish species” or “15 000 species occurring from fresh to brackish waters”. 

Response: 13,000 was used.

Comment: L51-L54: Move to the end of the sentence of L55 to read as follows: “Despite freshwater habitats only comprising of 1% of the Earth’s surface water, they support 43% (13 000 species) of the world’s freshwater fish [2,5], rich in biodiversity across all trophic levels [e.g. Vörösmarty et al., 2010; Carpenter et al., 2011]. As a result of human dependence on water, either for direct uses like irrigation and drinking or indirect uses like hydro-electrical energy production and food fishing, many freshwater fish species have become highly endangered [1, 6-9]” L57-L59: Remove sentence.

Responses: lines 51-59 were re-arranged and revised as recommended by the reviewer. References were added as requested.

Comment: L60-L62: Include a reference and change sentence to read as follows: “Effective management for the assessment of freshwater fish biodiversity and population status is therefore a priority, however, can be considered a great challenge.

Response: Text was revised as suggested, reference for this sentence was included.

Comment: L62-L65: I would not include reference number 8 (Lind et al., 2012) and number 10 (Leprieur et al., 2008) in your in-text citation for this sentence. Reasons being, the focus is still on a global point of view and you only mention fish invasions in the next sentence (L65-L66).

Response: The reference to Leprieur et al., 2008 was removed, however the reference to Lind et al., 2012 was kept as it does not refer specifically to fish invasions and is more general.

Comment: L65-L66: Change sentence to read as follows: “Further human derived pressure on natural fish diversity has also been documented due to introduction and invasion of exotic fish species”. Remove reference 12 (Goren and Ortal, 1999) and 14 (Ribeiro and Leunda, 2012) in your in-text citation, as again you are focusing on a global point of view here and focus specifically on Israel and the Mediterranean freshwater ecosystems in the next paragraph.

Response: Text was revised, and selected references were removed as suggested.

Comment: L67-L68: Insert the word “arid” before “climatic conditions.

Response: added.

Comments: L69-L77: 

Comment: Insert the catchment area size in brackets to read as follows: “biggest surface freshwater reservoir in the middle east (ca. 170 km2)”. 

Response: Catchment area size was added in brackets.

Comment: Change “are highly regulated” to “it is highly regulated”.

Response: Text was changed as suggested.

Comment: Include reference 12 (Goren and Ortal, 1999) in the in-text citation “[16,17]”.

Response: Reference was included. 

Comment: L71: Instead of using the term “exotic” I would opt for using “non-native”. Also insert the word “freshwater” before “fish”

Response: Comment was accepted, and text changed accordingly.

Comment: L72: Remove reference 12 and 18 from in-text citation as this sentence is more specifically mentioned in reference 19.

Response: Requested references were removed.

Comment: L72-L74: Include references for this sentence.

Response: A reference for this sentence was added

Comment: L74: When including the common and scientific name for the first time, first write out the common name followed by the scientific name with the original author/identifier in brackets. Change to read as follows: “The native cyprinid Kinneret bleak Mirogrex terraesanctae (Steinitz, 1952)”. 

Response: Changed here and throughout text that species are first mentioned properly. According to taxonomy conventions, the original identifier of the species was written in brackets when genus/species name were changed since first identified but without brackets when the original identification name was kept to date. 

Comment: L75-L76: Change sentence to read as follows: “From the native cichlids Galilee tilapia or St. Peter’s fish Sarotherodon galilaeus (Linnaeus, 1758) is the most abundant”. Also can you specify how they are abundant? By number or biomass as was done in the previous sentence for the cyprinid.

Response: Text was revised as follows: “Based on fishery landings, the native cichlid, Galilee tilapia, or St. Peter’s fish Sarotherodon galilaeus (Linnaeus, 1758), is abundant, most valuable and sought after."

Comment: L77: Scientific names are predominantly used over common names once the species has been introduced. Also, S. galilaeus African distribution is more centralised and western, therefore, please change sentence to read as follows: “While M. terraesanctae is endemic to Israel, S. galilaeus is widespread in central and western Africa”. Please remove reference 21 (Lemoalle and de Condappa, 2010) from the in-text citation for this sentence, as this reference does not refer to S. galilaeus African distribution which is of focus here but rather to the Volta basin (west coast of Africa).

Response: Text was changed as suggested and checked that scientific rather than common names are used throughout manuscript. Reference 21 was removed.

Comments L78-L82: 

Comment: Please rephrase sentence to read as follows: “Non-native species, which were unintentionally introduced, include the European eel Anguilla (Linnaeus, 1759) that hitchhiked with mullet fry caught in coastal estuaries and stocked into the lake, the hybrid tilapia Oreochromis niloticus (Linnaeus, 1758) x Oreochromis aureus (Steindachner, 1864) that escaped from aquaculture ponds and the peacock bass Cichla kelberi (Kullander and Ferreira, 2006) that was probably released in the lake by hobbyists [22]”.

Also, please include references for the introductions of the European eel and the hybrid tilapia in the in-text citation as currently you only have a reference for the peacock bass.

Response: Text was updated by relevant references and revised.

Comments L82-L83:

Comment: Here you listed Gambusia affinis as “exotic” stipulating that it is not harmful to native species, however, this is not the case as G. affinis is listed as one of the world’s worst non-native invasive species (Lowe et al., 2000). Please rephrase this sentence to read as follows: “Other non-native species, which were intentionally introduced and invasive include the mosquito fish Gambusia affinis (Baird and Girard, 1853) which were introduced to control mosquitoes. Please also include a reference for this sentence.

Response: The sentence was revised. A reference for the introduction of Gambusia fish was added

Comments L83-L91: Please rephrase this sentence to read as follows: “Nevertheless, significant non-native species which were introduced to enhance commercial fishing included the common carp Cyprinus carpio (Linnaeus, 1758), the silver carp Hypophthalmichthys molitrix (Valenciennes, 1844), the thin lip grey mullet Chelon ramada (Risso, 1827) and the flathead grey mullet Mugil cephalus (Linnaeus, 1758)”.

Response: Text was revised as suggested. 

Comment: Please note that the genus name “Liza” has been updated to “Chelon”. For assurity, please check that all species mentioned in this paper are up to date by reviewing them in the Catalogue of Fishes Database. 

Response: We checked all species names mentioned in this manuscript by reviewing them in the Catalogue of Fishes Database, to our knowledge all species names are currently up to date.

Comment: Also, please include a reference for this sentence.

Response: References for introduction of non-native species mentioned here were added.

Comment: L86-L88: Please include a reference for this sentence.

Response: A reference for this sentence was added.

Comment: L90-L91: Change “estimate” to “estimates” and change “Kinneret bleak” to “M. terraesanctae”.

Response: Text was changed as suggested.

Comment: L93-L94: Rephrase sentence to read as follows: “In recent years, DNA polymorphism, a more reliable and accurate method, is increasingly used for taxonomic and ecological research”.

Response: Comment was accepted, and text changed. 

Comment: Please include a references for this sentence by providing other examples of past literature where DNA polymorphism has been used successfully.

Response: A few example references were added to text.

Comment: L94: Change “sequence” to “sequences”.

Response: Comment was accepted, and text changed. 

Comment: L95: Remove “species” before “DNA barcoding”.

Response: Comment was accepted, and text changed. 

Comment: L98: After “environmental water samples, include “known as E-barcoding”.

Response: Comment was accepted, and text changed.

Comment: L98-L99: Elaborate more on what reference databases have been used and are currently being used successfully, e.g. BOLD. Explain the process in more detail. This will strengthen your argument and lead nicely on to your aim for the study.

Response: The process of molecular identification of specimen was explained in more detail, and the Barcode of Life Data systems was presented and described shortly.

Comment: L101: Change “question” to “questions”.

Response: Comment was accepted, and text changed.

Methods

The experiments or intervention used was DNA barcoding. DNA barcoding was appropriate for addressing the research question. However, the data collected was not interpreted accurately. I found it very hard to follow the protocols and feel that another researcher would battle to reproduce this study based on the methodology reported. For instance, sampling methods are not clear, number of samples collected for each species per site are not clear as well as how many successful sequences were used from this sample size. No mention of whether the fish collected were stored in formalin for taxonomic identification as well as no mention of photos of samples for each species were taken to upload with BOLD. References were not provided or corresponded to the authors point. Data analysis section needs work. Please could the authors rectify the above and include them into the materials and methods section.

Response: see below a point-by-point response.

Comment: L106-L108: Firstly, I would not recommend saying “in the context”, rather say “from the Sea of Galilee and surrounding areas”. Secondly, what is this extensive sampling scheme that was carried out to obtain samples from each species that could be found in the Lake? For instance, how were these fish collected? Electrofishing? Seine netting? It is important to incorporate this detail here in the beginning of your materials and methods section as well as the years in which this was done (2014 and 2015?) to correspond to your permit numbers.

Response: Text was changed as suggested. Details regarding the sampling scheme are explained in more detail in the following lines as suggested and addressed in the next comment. Information on the permits for sampling was added in the ethical approval section.

Comments: L108-L113: What do you mean by “12 sampling attempts” and “few additional attempts”? This gives me the impression that collection was done on 12 separate occasions but nothing was collected by using the word ‘’attempts”. Also, Borovski et al. (2018), does not mention anything about “12 sampling attempts” but rather provides more detail in a table and figure of the location of the sites sampled from. I would recommend changing this sentence to read as follows: “The Sea of Galilee, the main sampling site, and surrounding sites such as Asi stream (one of Beit She’an streams, south to the lake and part of the eastern watershed of Israel), Ein Afek pools (close to Acre, i.e. Akko, part of the western watershed of Israel) and Ein Te’o spring (close to Hula Lake, northern to Lake of Galilee and part of the eastern watershed of Israel) (Fig. 1) was sampled during 2014 and 2015 to collect samples of species by means of “seine netting” (whichever method you used for collection) that predominantly occur in this region of Israel for DNA barcoding. Numbers of individual samples collected for each site and species are recorded in Table 1”. 

I would also recommend including a map (Fig.1) of your sites including all the small town names and streams as mentioned above in brackets on the map for a point of reference and in Table 1 include columns labelled “site” (Sea of Galilee/Kinneret, Asi stream, Ein Afek pools, and Ein Te’o spring) and “sample size” indicating the number of samples collected for each species for each site. 

Response: Sampling was done on 12 different occasions, all ‘attempts’ yielded fish for sampling. Details about sampling method, locations and yields were explained in more detail based on the recommendation for text revision given here. Additionally, a new Table 1 was added to provide details about number of fish sampled compared to number successfully barcoded from each species and from each location. Instead of providing a map as suggested we refer to figure 1 in Borovski et al. (2018) were this data was already published, as we would like to avoid publishing the same data twice. We however, modified the map in Figure 2 to include sampling site names so this information is readily available in this manuscript as well.

Comment: Within table 1 change the column “N samples” to “N samples barcoded”. This will give us a clear indication of the number of samples per species per site that were successfully barcoded from the sample size collected for each species. 

Response: Within the table (now table 2) ‘N samples’ was changed to ‘N barcoded samples’.

Comment: Be consistent of your labelling and mentioning of Sea of Galilee rather than “Lake Kinneret” as this was in brackets in the introduction section (L69).

Response: Text was reviewed throughout to ensure that ‘Sea of Galilee’ is used rather than ‘Lake Kinneret’.

Comment: L113-L114: When referring to fish being taxonomically identified to species level, were they identified in the field and returned to the environment or where they sacrificed and taken for identification under lab conditions (preserved entire sample in formalin)? Please specify in this sentence. Also, under DNA barcoding methods of collection, it is important to photograph each sample and upload this photograph on the BOLD database with it its barcode index number (BIN number), was this done? If so, please include this information. Furthermore, why specifically fin clips and not a muscle tissue sample? According to Weight et al. (2012) (DNA barcoding fishes, a good reference to include here), even though fin clips work, the most effective material is a small muscle biopsy from the specimen taken from the right side of the body. Please explain in detail why a fin clip over muscle tissue was done. Lastly, if fin clips were taken, was there evidence of the fish surviving if returned to its habitat and not preserved? Were there any infections caused which may have made the individual more susceptible to predation?

Response: Text was revised to include more information about sampling procedures and fish handling. Text is now informative with regards to all these points. In most cases fish were returned to their environment after identification in the field and sampling of a fin clip. We have experience with fish (including critically endangered fish) in lab conditions showing that after recovery from anaesthesia, fish return to normal behaviour and their survival is mostly unaffected. As fins easily regenerate and fish integument is unharmed, fin clip samples were chosen over muscle and in our experience, fin clips prove to be effective for DNA isolation and barcoding. Some of the fish were sacrificed photographed and deposited in the National Natural History Collections of the Hebrew University, in the new Table 1, the information of number of specimens that were deposited and their Museum catalogue numbers was added. This data was also made available in BOLD.

Comment L118: Change to read as follows: “DNA extraction, polymerase chain reaction (PCR), purification and sequencing”. Combine your original sub-headings of “DNA extraction” and “PCR products and sequencing of PCR products” into one paragraph.

Response: The two sub-sections were combined under the sub-heading of “DNA extraction, polymerase chain reaction (PCR), purification and sequencing”.

Response: done as suggested.

Comment L119-L120: For the salting out method, a reference, Martinez et al. (1998) is provided, however, this citation refers to the salting out method from fish whole blood. Ideally, it would be better to provide a reference that directly links this method for extracting DNA from fin clips of fish as was stipulated, for example Taggart et al. (1992) or Sunnucks and Hales (1996) which in most cases for other references, related to this method, modified it from Miller et al. (1988).

Response: A reference for the salting out method from fin samples was added.

Comment L120-L126: Delete. Not necessary to go into detail if you provide the correct reference for the salting out method and protocol in which you followed. 

Response: These lines were deleted.

Comment L132-L135: The primers indicated in the manuscript are not the same primers used as Ward et al. (2005). They used FishF1, FishF2, FishR1, FishR2 and the primers included in the manuscript were coxla_fw, coxla_rv. Hence, please do not indicate that they were “adopted” from Ward et al. (2005) but rather provide a reference that used the same primers as was indicated in this manuscript. Change sentence to read as follows: “A partial fragment of (include the number of base pairs here) of the mitochondrial cytochrome c oxidase subunit 1 gene (COI) was amplified by PCR using the primers …….”.

Response: We used primers FishF1 and FishR1 developed by Ward et al. (2005) under the names cox1a_fw and rv, which were internal names used in our lab. The sequence, however, is identical. We therefore changed the primer names in the text to match the original names from Ward et al. Additionally text was revised according to the comment.

Comment L138: Please include the amount of water used to complete the 20μl reaction volume and change “PCR profile” to “PCR thermal protocol”.

Response: We added the amount of water used in our PCR protocol to complete to 20 μl, and changed “PCR profile” to “PCR thermal protocol” as suggested.

Comment L139-L142: Please include the reference that you used this protocol from or modified it from.

Response: We developed a thermal protocol in our lab that was effective for amplification of products by these primers and described it in the text, therefore no reference was provided.

Comment L144-L146: Change sentence to read as follows: “Successful PCR products were purified using an Exonuclease I-Shrimp Alkaline Phosphate (ExoSAP-IT) method (USB, Cleveland, OH) and sequenced using BigDye V3.1 (add manufacturers name) terminator chemistry in the forward and reverse direction on an ABI PRISM 3730xl DNA analyser (add manufacturers name) following the manufacturers protocol”.

Response: Text was revised as suggested and manufacturers of materials and instruments were provided where they were missing.

Comments: L148:L159

For the data analysis section, methodology processes are not clearly explained and a lot more work needs to be done here. For instance, under this section it needs to be mentioned that successful sequences for each species was deposited into the Barcode of Life Data (BOLD) System (Ratnasingham and Hebert, 2007; https://www.boldsystems.org) under the project name LKCOX (remove this sentence from results section under molecular barcoding, and add it under data analysis).

Response: Text was revised as requested, the relevant sentence was moved here from the results section, and blended with the existing revised text.

Comment: In addition, once deposited, what exactly was the role of the BOLD database? Were the sequences from this study used to compare against available reference database sequences to find the closest/nearest matching species which was displayed as a percentage probability (0-100%)? Also, for each species, what was the number of sequences submitted into BOLD for comparison (is this the N samples column in Table 1)? Next, were the publicly available sequences with a >98% similarity/match downloaded from the BOLD reference database (and how many) to include with your sequences in running the haplotype analysis. In the results section, L196-L197 includes the above information, move this sentence here to the data analysis section. Also, L204-L206 of the results section needs to be moved here to the data analysis section.

Response: This section was rearranged, including the requested lines from the results section, as suggested. Now, the text includes all requested information. Regarding the number of sequences submitted for each species – this is described in Table 1 and referred to here.

Comment: How did you run the haplotype (phylogenetic) analysis? How did you get the relative number of haplotypes for each species? Did you use the software DnaSP? 

Response: After alignment of sequences, we manually inspected polymorphisms and named polymorphic sequences of the same species as different haplotypes. This is now stated more clearly in the text. 

Comment: Was the neighbour joining tree rooted or unrooted for both Figure 1 and Figure 2? Please provide a reference for the bootstrap method of 1000 replicates.

Response: Trees in both figures were unrooted, this is now stated in the methods section and also in both figure legends. A reference for the bootstrap method was provided.

Comment: How was average pairwise haplotype distances calculated? Did you use ARLEQUIN software? 

Response: The average pairwise haplotype distances was calculated by summing up all relevant pairwise K2P distances, calculated using MEGA software (described in the text), and dividing them by the number of comparisons.

Comment: Please stipulate what version of the BioEdit software was used and indicate what the total length of base pairs was used or trimmed to by the software.

Response: This data is now added in the relevant places in the text.

Comment: Considering relevant species were sampled from four sites (mainly the Sea of Galilee), why was a haplotype network not included to show if some species indeed differed according to site based on their haplotype groupings?

Response: This suggestion applies mainly to the Garra complex, for which we analyzed samples from different locations showing different haplotypes. Most species have different haplotypes within the lake population (including Garra), leaving such an analysis less interesting. Nevertheless, this point for Garra, including from other countries, in our opinion, is presented in a more specific and legible way than a haplotype network in Figure 2.

Results

Results are very interesting! Figures and tables are clear and readable, however Figure 2 is not of great quality and more information needs to be incorporated in to the map (Fig 2A; see below corrections for further recommendations). Table 2 legend should be above the table and not below it. 

Response: see below a point-by-point response.

Corrections:

Comment: I would recommend for Figure 1 to exclude the K2P distances combined here and provide them as separate tables as has been done for Table 2, just separate it according to family using sub-headings. 

Response: We prefer to leave the K2P distances within the figure, instead of separating them in three tables. Separate tables will be small and require scrolling among them and back in reference to Fig. 1, which, we think, will be less convenient to the reader. This way the distances and phylogeny can be viewed simultaneously to demonstrate better the points of interest.

Comment: Also correct acronyms for Figure 2 and Table 2 (see below corrections for further recommendations). 

Response: This was corrected as per the below comments.

Comment: L169-L170: Delete.

Response: Deleted.

Comment: L171: Are there 25 fish species reported in the Lake? From the in text references reported for this sentence, Goren and Ortal (1999) reported 19 native species in the Lake; Ostrovsky et al. (2014) reported 19 native species in the lake; and I did not come across the number of native species for Ostrovsky et al. (2014) fisheries management chapter. Even Borovski et al. (2018) reports 19 native species and 8 exotic species, which gives a total of 27 and not 25. Please clarify.

Response: In Ostrovsky et al. (2014) there are 18 native species in the lake (19 stated, probably after Goren and Ortal, however, in the table actually detailing the lake species (table 16.1 in Ostrovsky et al., 2014), only 18 species are listed). In comparison to Goren and Ortal (1999), the missing species is Nun galilaeus, which was native to the Hula valley and suspected to have migrated into the lake, but anyways, since the 1950's this species is considered extinct. Therefore, we updated our introduction to report 18 native fish species and changed text here to be more specific by reporting how many of the native species we obtained barcode sequences for, and how many non-native species.

Comment: L172-L176: Please provide the original author/identifier in brackets for all species mentioned in these lines. Change “schemes” to “sites”.

Response: author/identifiers were added to each species as appropriate.

Comment: L177: At the end of the sentence add Table 1 in brackets.

Response: Added.

Comment: L177-L179: Rephrase this sentence to read as follows: “Additionally, 20 sequences were analysed from two fish species, H. molitrix (19 sequences) and O. aureus (1 sequence), that were suspected as hybrids with aquaculture fish (hereafter referred to as ‘suspected hybrids’) (Table 1)”.

Response: Text was revised as suggested.

Comment: L179-L180: Delete and move to data analysis section under materials and methods.

Response: This sentence was moved to m&m, data analysis section.

Comment: L181-L182: As previously mentioned, can the authors make clear how many specimens were collected for each species, and out of this sample size, how many were successfully sequenced and add this to Table 1 as some sequences may have been unsuccessful. Also, at the end of this sentence add Table 1 in brackets.

Response: The new Table 1 includes the requested information.

Comment: L183-L184: At the end of the sentence add Table 1 in brackets.

Response: Added, but now Table 2.

Comment: L186: Add the “number” of species where new barcodes were added to existing barcodes in BOLD.

Response: The number of species with novel barcodes was already in the sentence, however, to clarify, we changed the sentence to read as follows: “For ten species with barcodes existing in BOLD, 12 new barcodes were added. These added haplotypes were always those with the lower frequency in our data (Table 2)."

Comment: L188-L190: Change Table 1 legend to include sites and sample sizes for each species.

Response: This information is now detailed in the new Table 1.

Comment: L196-L197: Delete and move to data analysis section under materials and methods.

Response: This sentence was deleted here and moved to data analysis section of methods.

Comment: L197-L199: Specify the exact number of species where molecular identification agreed with the original morphological identification.

Response: We added that for 19 out of 22 molecular identification agreed with the original morphological identification.

Comment: L199-L201: Delete and move to data analysis section under materials and methods.

Response: This sentence was deleted here and moved to data analysis section of methods.

Comment: L204-L206: Mention how many species were in disagreement. Please rephrase the sentence to read as follows: “Three species were found to have disagreements between morphological and molecular barcoding identification. Further investigation was carried using all available COI barcode records from BOLD of query and related species”.

Response: Text was revised as suggested, and the methods part was moved into data analysis section of methods.

Comment: L206: Cannot start a sentence with the genus name of a scientific name abbreviated, need to write it out in full, “Acanthobrama”. Delete “we”.

Response: Text was revised as suggested.

Comment: L211-L214: Remove from results section and add to discussion section. You are only stating the main points of results and should not discuss them in depth in this section.

Response: Text was revised as suggested.

Comment: L215-L217: Please rephrase the sentence to read as follows: “Three Gambusia affinis samples collected, all with the same haplotype (GaA), was identified by BOLD as G. holbrooki (100% fit), followed by G. affinis (99.8% fit).

Response: Text was revised as suggested.

Comment: L217-L220: Please delete the last part of this sentence and move to discussion section. Please rephrase sentence to read as follows: “Comparison to all available G. holbrooki and G. affinis records from BOLD indicated that GaA haplotype clusters together with G. holbrooki records”.

Response: the last part of the sentence was deleted and moved to the discussion section.

Comment: L222: Remove “identification module”.

Response: deleted.

Comment: L223-L224: Rephrase this sentence to read as follows: “This disagreement will be discussed further later under its own sub-section”.

Response: Text was revised as suggested.

Comment: L225-L226: Rephrase sentence to read as follows: “Molecular identification using BOLD for samples of fish suspected as hybrids provided genetic evidence to support hybridization. 

Response: Text was revised as suggested.

Comment: L228: Change “into” to “in”.

Response: Text was revised as suggested.

Comment: L229-L231: Delete this sentence, starting from “This is consistent”.

Response: Text was revised as suggested.

Comment: L234-L235: Add a full stop after “in both directions” and delete the remainder of the sentence starting from “again”.

Response: Text was revised as suggested.

Comment: L236: Change “another” to “further”.

Response: Text was revised as suggested.

Comment: L239-L252: For genetic distance from barcoding data, I would start off with discussing the results from the neighbour-joining phylogenetic tree and incorporating the K2P genetic distances with this.

Response: This paragraph begins with a general statement about the scale of the K2P distances, followed by a general description of the phylogenetic tree. In the next paragraph, a more detailed description of the interesting results in the tree are combined with the K2P genetic distances. We believe this is a reasonable and logical way to present the results. 

Comment: In figure 1, for certain haplotypes there as asterisk next to them. In the Figure 1 legend, please indicate what these asterisks mean. 

Response: Description of the asterisks meanings were added to figure 1 legend.

Comment: L260-L263: Please rephrase this sentence to read as follows: “Unexpected results were, however, found within the Cichlidae clade of the neighbour-joining phylogenetic tree. Oreochromis aureus (Haplotype OaA) clustered together with Sarotherod galilaeus (Haplotype SgA), a species belonging to a different genus, rather than with O. niloticus (Haplotype OaB) with which its genus is shared”.

Response: Text was revised as suggested.

Comment: L263-L267: Please rephrase this sentence to read as follows: “Support from K2P distances between O. aureus and S. galilaueus (0.0095) is smaller in comparison to distances between other cichlids of Lake Galilee belonging to different genera, and even smaller compared to cyprinids belonging to different genera (0.08-0.23) (Fig. 1)”.

Response: Text was revised as suggested.

Comment: L267-L268: Delete this sentence here and move it to discussion section.

Response: This suggestion was followed.

Comment: L269: Change “G. rufa” to “Garra rufa”.

Response: Text was revised as suggested.

Comment: L270: delete “we find”.

Response: Deleted.

Comment: L271-L273: Please rephrase this sentence to read as follows: “Identical K2P distances was, however, found between Acanthobrama telavivensis and Mirogrex terraesanctae, formerly known as A. terraesanctae and later moved to a separate genus”. Reference [36] has been removed here as in the result section you do not include in text references.

Response: This suggestion was followed.

Comment: L273-L274: Please delete this sentence, starting from “This relatively” and move to the discussion section.

Response: This suggestion was followed.

Comment: L277: Delete “As mentioned before”. Start sentence with “Comparison of Garra rufa haplotypes”.

Response: Text was revised as suggested.

Comment: L281-L282: Delete “Samples identified morphologically as”. Start sentence with “Garra rufa fish”.

Response: Text was revised as suggested.

Comment: L283: After Ein Te’o put “north of the Sea of Galilee” in brackets. 

Response: Text was revised as suggested.

Comment: For Figure 2A, can the authors please explicitly label Asi stream, Ein Afek and Ein Te’ o stream in text in the correct location on the map to make these locations clearer in relation to the haplotypes. Also, include labels for eastern and western watersheds of Israel on the map (Fig. 2A). Furthermore, can you give Haplotype GrD its own circle and colour and make GrA haplotype in the Sea of Galilee the same colour as in Ein Te’o (the colours on the map look different).

Response: Figure 2 was edited, site names were included (except for Ein Afeq, since no Garra fish were sampled there), eastern and western watersheds were not marked on the map since all sampled fish are from the eastern watershed of Israel. Haplotype GrD received its own circle and colour and we made sure that GrA haplotype in the Sea of Galilee is the same colour as GrA in Ein Te’o. Furthermore, map resolution was improved.

Comment: L284: Make “the” prior to Sea of Galilee samples capitalized.

Response: Text was revised as suggested.

Comment: L284-L287: Please rephrase this sentence to read as follows: “The Sea of Galilee samples had five fish from Haplotype GrA, two fish from Haplotype GrC and one fish from Haplotype GrD. All three fish from Ein Te’o shared Haplotype GrA and all three fish from Asi stream shared Haplotype GrB (Fig. 2A).

Response: Text was revised as suggested.

Comment: L289-L298: Please rephrase Figure 2 legend to read as follows: “Sampling sites and clustering of Garra spp. Samples based on COI haplotypes. Acronyms of populations are: GrA – Garra rufa from Haplotype A; GrB – Garra rufa from Haplotype B; GrC – Garra rufa from Haplotype C; GrD – Garra rufa from Haplotype D; Gg – Garra ghorensis from the southern basin of the Dead Sea; Gj – Garra jordanica from the northern basin of the Dead Sea; GnA – Garra nana from Haplotype A; GnB – Garra nana from Haplotype B. (A) Map showing the sampling sites in a regional context. Different Garra spp. Haplotypes were denoted by differently coloured circles in each sampling site on the map of Israel, Jordan and Syria regions. (B) A dendrogram constructed for Haplotypes obtained from samples collected in this study alongside Haplotypes mined from BOLD (see Process ID’s in Table S1). Next to the edges, haplotype acronyms are given as indicated above. Numbers next to edges denote percent bootstrap support. Colours of Haplotype text match the colour of the circles on the map (A)”.

Response: Text of the figure legend was revised as suggested.

Comment: L300: Delete “To put into context the Garra spp. Sampled from Israel,”. Start sentence with “Sequences of 16 Garra spp. From”.

Response: Text was revised as suggested.

Comment: L301: Delete “a” before “geographic location”.

Response: Deleted.

Comment: L302: Add “assigned” before “to them” and add “see” before “Table S1” in brackets. Add “neighbour-joining” before “phylogenetic tree (Fig 2B)”.

Response: Text was revised as suggested.

Comment: L305: After 0.0381 in brackets, add “; Table 2). After “were found in fish from” add “the Sea of Galilee and a northern location in Syria (Fig 2A). Delete “two isolated locations” and “GnA, the G. nana haplotype found in the Sea of Galilee was found also in samples from a northern location in Syria”.

Response: These lines were rephrased to read as follows ‘The two haplotypes of G. nana (GnA and GnB) were relatively distinct from one another (K2P = 0.0381; Table 3). GnA, was found in fish from the Sea of Galilee and the more northern location in Syria, while GnB, was found in fish from an isolated location more southern in Syria (Fig 2A).’.

Comment: L308-L311: Please rephrase this sentence to read as follows: “Israel G. rufa samples from the Sea of Galilee and Ein Te’o had Haplotypes GrA, GrC and GrD. These Israel G. rufa Haplotypes clustered with BOLD Haplotypes of G. jordanica (Haplotype Gj), fish sampled from streams in the northern basin of the Dead Sea in Jordan (Fig. 2B)”.

Response: Text was revised as suggested.

Comment: L311-L314: Please rephrase this sentence to read as follows: “Close by, yet separated, in the phylogeny, was G. rufa fish, Haplotype GrB, from Asi stream, suggesting again greater similarity to G. jordanica, but some genetic separation from the Sea of Galilee population”.

Response: Text was revised as suggested.

Comment: L318: Insert “GrAsia –“ before “Turkey, Iran and Iraq” in the brackets.

Response: Inserted.

Comment: L318-L320: Delete this sentence starting from “Thus, the results indicate that” and move it to the discussion session.

Response: Deleted and moved.

Comment: L321-L327: Figure legend for Table 2 should go above the table and not below the table. Provide a reason as to why Gr North has an asterisk sign and the other locations do not. Change acronyms for populations to read as follows: “Gr North – Garra rufa from Sea of Galilee and Ein Te’o of northern Jordan valley; GrAsi – Garra rufa from Asi stream of Beit she’an valley; Gj – Garra jordanica from the northern basin of the Dead Sea; Gg – Garra ghorensis from the southern basin of the Dead Sea; Gr Asia – Garra rufa from Asian countries including Turkey, Iran and Iraq; GnA – Garra nana from Haplotype A; GnB – Garra nana from Haplotype B”.

Response: Figure legend for this table (now Table 3) was placed above the table and revised as suggested.

Comment: L333-L337: Please rephrase this sentence to read as follows: “The Israeli G. rufa fish (Gr North) are somewhat genetically divergent from G. rufa fish from eastern and norther Asian countries (Gr North vs Gr Asia mean K2P distances = 0.0025 vs 0.0462; Table 2)”.

Response: To clarify the point of interest, the text was revised to read as follows: ‘The Israeli G. rufa fish, are genetically divergent from G. rufa fish from farther eastern and northern Asian countries (Mean K2P between Gr North and Gr Asia = 0.0462, and between Gr Asi and Gr Asia = 0.0469; Table 3).". The text following these lines was also revised similarly to improve clarity.

Comment: L340-L341: Delete “from the Northern Jordan valley” and replace it with “(Gr North)”.

Response: Text was revised as suggested.

Comment: L343: Insert “south Asi stream of” before “Beit she’an valley”. Delete “(Asi stream)” and replace it with “(Gr Asi)”.

Response: Text was revised as suggested.

Comment: L344-L345: Delete “in the Hula valley”.

Response: Text was revised as suggested.

Comment: L350: insert “(mean K2P of 0.0587) after “from Asia”.

Response: Text was revised as suggested.

Comment: L351: Rephrase to read as follows: “G. rufa (mean K2P of 0.014) (Table 2)”.

Response: Text was revised as suggested, mean K2P should be 0.0414, we assume this was the reviewer's intention.

Discussion

The discussion section is almost a repetition of the authors results section. More evidence needs to be provided here to support your results. Elaborate by including more examples of relevant recent literature to strengthen your arguments. Results are interesting, however a lot more work needs to be done in this section. Also, it is important to include any limitations that were raised during this this study and should be highlighted and discussed here by providing ways in the future as to how it can be avoided or improved on. Also, there is no conclusion section, which is very important to drive your take home message for this study. For instance, in the conclusion section by adding a how this study could contribute to the conservation of Israel’s native species and identification of hybrid and non-native invasive species through DNA barcoding would be a good as well as how it will improve in the future from this perspective.

Furthermore, before discussing the G. rufa complex in the discussion, authors need to discuss the other disagreements that were found from your results, namely A. lissneri and A. telavivensis; G. holbrooki and G. affinis; hybrids; Cichlidae; and Acanthobrama-Mirogrex, in more detail by referring to or by providing recent, relevant and supportive literature. In the results section, I indicated which sentences need to be moved to the discussion section relative to the above sentence as they should not be present in the results section.

Response: the discussion was considerably rewritten incorporating all the suggestions of the reviewer including a conclusions section. Also, see below a point-by-point response to specific comments.

Corrections:

Comment: L361: Reference [25], i.e. Ward et al. (2005) does not directly refer to DNA barcoding as being an important part of BOLD, perhaps rather refer to reference [40].

Response: Sentence was changed to emphasize the utility of fish DNA barcoding including relevant references. 

Comment: L361-L362: Please rephrase this sentence to read as follows: “In a previous study conducted by [39], the cichlids of Israel were barcoded with success”.

Response: Text was rephrased as suggested.

Comment: L362-L363: Please rephrase this sentence to read as follows: “In this study, focus was given to the fish found in the Sea of Galilee and surrounding water bodies in the region, including species other than cichlids”.

Response: Text was rephrased as suggested.

Comment: L367: After “12 species”, add “mostly belonging to the Cyprinidae and Cichlidae,”.

Response: Text was rephrased as suggested.

Comment: L370: For references [24,40] did both studies focus on similar families as this study? Is this a fair comparison?

Response: References of studies on similar species in which similar distances were found were added.

Comment: L375-L377: Please provide a reference for this sentence.

Response: References for this sentence were added

Comment: L389: Please indicate how recent this isolation occurred in by means of a time frame.

Response: We added the dating of this isolation (~24,000 years ago) to the text, along with a supporting reference.

---

## [Decision Letter · Decision Letter 1]

16 Mar 2022

PONE-D-20-39138R1Establishing and using a genetic database for resolving identification of fish species in the Sea of Galilee, IsraelPLOS ONE

Dear Dr. David,

Thank you for submitting your manuscript to PLOS ONE. After careful consideration, we feel that it has merit but does not fully meet PLOS ONE’s publication criteria as it currently stands. Therefore, we invite you to submit a revised version of the manuscript that addresses the points raised during the review process.

Both outside reviewers suggested minor revisions. Once these revisions are completed then the manuscript will be suitable for publication. I will not send the manuscript out again for review but will review the manuscript in the context of the reviewer comments.

We look forward to receiving your revised manuscript.

Kind regards,

Mary Peacock, Ph.D.

Academic Editor

PLOS ONE

Journal Requirements:

Reviewers' comments:

Reviewer's Responses to Questions

**Comments to the Author**

1. If the authors have adequately addressed your comments raised in a previous round of review and you feel that this manuscript is now acceptable for publication, you may indicate that here to bypass the “Comments to the Author” section, enter your conflict of interest statement in the “Confidential to Editor” section, and submit your "Accept" recommendation.

Reviewer #1: All comments have been addressed

Reviewer #2: All comments have been addressed

2. Is the manuscript technically sound, and do the data support the conclusions?

Reviewer #1: Yes

Reviewer #2: Yes

3. Has the statistical analysis been performed appropriately and rigorously? 

Reviewer #1: Yes

Reviewer #2: Yes

4. Have the authors made all data underlying the findings in their manuscript fully available?

Reviewer #1: Yes

Reviewer #2: Yes

5. Is the manuscript presented in an intelligible fashion and written in standard English?

Reviewer #1: Yes

Reviewer #2: Yes

6. Review Comments to the Author

Reviewer #1: The paper was really improved after revision but some minor points still need some corrections. These points were signed in the text.

Reviewer #2: Introduction

Comment: L8: Change “derived pressure” to “derived pressures”.

Comment: After “Further human derived pressures on natural fish diversity has also been documented due to introduction and invasion of exotic fish species [10-13].” continue with the following sentence, “Conservation measures intended to mitigate the impact of human derived pressures have largely been slow and inadequate, and as a result, populations of many freshwater species have been declining rapidly [5, 6, 8, 15].”, then end the paragraph with “Assessment of freshwater biodiversity and population status for effective management is, therefore a priority, however, can be considered a great challenge [1, 6, 14, 15].”

Comment: Start “Due to its arid climatic conditions ……” as a new paragraph.

Comment: L19: Change “about” to “approximately”.

Comment: L22-L24: Please change sentence to read as follows: “The native cyprinid Kinneret bleak Mirogrex terraesanctae (Steinitz, 1952), which is endemic to the sea of Galilee, is probably the most abundant species in terms of both numbers and biomass [20].”

Comment: L24-L26: Please change sentence to read as follows: “The native cichlid, Galilee tilapia, St. peter’s fish or Mango tilapia Sarotherodon galilaeus (Linnaeus, 1758), which is widespread in central and western Africa [8], is also abundant in terms of numbers and biomass, based on fishery landings, and is considered an important species to local fisheries [25].

Comment: L26-L27: Delete the sentence “While M. terraesanctae is endemic, S. galilaeus is widespread in central and western Africa [8]”.

Comment: L27-L31: Please change sentence to read as follows: “Non-native species, which were unintentionally introduced, include the European eel Anguilla anguilla (Linnaeus, 1759) that hitchhiked with mullet fry caught in coastal estuaries and stocked in the Sea of Galilee [21], the green swordtail Xiphophorus helleri (Heckel, 1848) [22] and the Peacock bass Cichla kelberi (Kullander and Ferreira, 2006), probably released into the Sea of Galilee by hobbyists [23]”. Out of curiosity, why did you replace X. helleri (spelling is with one “I” at the end, not two “I’s”) with the Oreochromis hybrid? Is it because the Oreochromis hybrid is not an exotic species and this was your focus? Would still be good to include the Oreochromis hybrid example in this paragraph.

Comment: L33-L36: Please change sentence to read as follows: “Additionally, significant non-native species including the common carp Cyprinus carpio Linnaeus 1758, the silver carp Hypophthalmichthys molitrix (Valenciennes, 1844), the thinlip grey mullet Chelon ramada (Risso, 1827) and the flathead grey mullet Mugil cephalus Linnaeus, 1758 were introduced to enhance commercial fishing [16, 22].”

Comment: L45: rewrite as “[e.g. 26-28].

Comment: L59: Replace “Here” with “For this study”.

Materials and methods

Comment: L75: Please include the dosage used for anaesthesia.

Comment: L77: Please change “100% ethanol” to “99.9% ethanol”.

Comment: L77-L78: Please change sentence to read as follows: “In most cases, after identification and sampling, fish recovered from anaesthesia and were returned to the environment after a monitoring period of ……. (please include this)”.

Comment: L78-L81: Please change sentence to read as follows: “Some fish, however, were sacrificed, photographed and fixed in formalin to be deposited in the National Natural History Collections of the Hebrew University of Jerusalem, Israel with a museum number (Table 1), serving as voucher specimens (transferred to 99.9% ethanol)”.

Comment: L79: Why were only some fish photographed and not all fish collected for fin clips? To my understanding, for DNA barcoding protocols, all individuals that were fin clipped should have photographs to be uploaded to the BOLD database. Also, why for so many of your species collected were no individuals taken as voucher specimens/deposited into the National collection? Is it because there are already these species present in the National Collection? I would expect for field sampling records in a study, at least 2 individuals of each species should be accessioned into the Natural collection. This allows other researchers to compare across years and study sites where a particular species is distributed.

Comments: Table 1:

• Please rephrase table heading to, “Fish species collected, barcoded and deposited at the National Natural History Collections of the Hebrew University of Jerusalem (NNHH-HUJ) by species and location”.

• Can you add the columns “Order” and “Family” in Table 1 to allow for the same format in Table 2?

• Can asterisks be added to the hybrids and a cross to A. lissneri as done in Table 2.

• Relabel “N samples barcoded” to “N successfully barcoded samples”

• Incorrect numbers of successful barcodes are reported in table 1 for A. flaviijosephi (9 instead of 8 for Beit She’an), G. rufa (7 instead of 8 for Sea of Galilee), H. molitrix, H. nobilis, and S. galilaeus when compared to the numbers of samples present in the LKCOX project on the BOLD database. For S. galilaeus, in the table you recorded 17 from the Sea of Galilee when in the BOLD database there are 14 records from this site; and for Ein Afek you recorded 4 in the table when there are 2 for this site. Also, on the BOLD database there are other sites including Ginosar Research (4), Naman River (2), and Volta Lake (4), could you have added some of these samples by mistake to the Sea of Galilee and Ein Afek samples?

• Double check the number of 176 published records is correct, in relation to above statement, and explain the difference between this number and 217 published records.

• Garra jordanica is not included in the table but is present in the LKCOX project on the BOLD database

• To my understanding, when depositing vouchers into a National collection, each individual gets a museum accession number. Why is there only one accession number for A. telavivensis when three specimens were deposited? The same for G. rufa, A. flaviijosephi, C. zillii, S. galilaeus and O. aureus.

• IS Cyprinus carpio carpio a sub-species of Cyprinus carpio? According to Eschmeyers Catalog of Fishes, there is no Cyprinus carpio carpio listed. Is this meant to be Cyprinus carpio?

Results

Comment: Double check the number of CO1 sequences (176) from the 22 freshwater species, does not add up to the 217 published records on the BOLD database.

Comment: Table 2: Change “N barcoded samples” to “N successfully barcoded samples”. Check the numbers reported under this heading are correct and correlate to the BOLD database published records.

Comment: Figure 2: Can you make GrA a darker purple. Also, can you make GrD a different colour like pink or yellow, currently the blue colour used is too similar to GrC.

Discussion

Thank you for including some of the limitations to your study as well as a conclusion section. This section brings the findings concisely together.

Comment: Please check that 176 CO1 sequences is correct from the 22 freshwater fish species.

Comment: Please rephrase the following to: “Thus, minor frequency new barcodes were identified for a high proportion of the native species, an indication for the genetic divergence and uniqueness of these regional native populations, except for a few discrepancies which will be discussed later on”.

Comment: Please rephrase the following sentence to: “For 19 out of the 22 species, morphological and DNA barcoding agreed on the species definition”.

7. PLOS authors have the option to publish the peer review history of their article (what does this mean?). If published, this will include your full peer review and any attached files.

Reviewer #1: **Yes: **Claudio de Oliveira

Reviewer #2: No

---

## [Author Response · Author response to Decision Letter 1]

31 Mar 2022

Response to reviewer’s comments:

Firstly, we thank again for the constructive comments on the revised version. We addressed all comments, as detailed below, and submitted a revised version integrating all these modifications. Reading over the revised version, indeed reads better and therefore, we believe it is ready to be accepted.

Reviewer #1

Responding to comments made in-text.

Comment: In the abstract the word ‘probably’ was highlighted in the sentence: “Notably, the cyprinid fish defined as Garra rufa, should be considered as Garra jordanica.”. 

Response: There was no comment attached, however, we decided to omit the word ‘probably’ from the revised text.

Comment: 8% is very high! (Referring to the sentence – “A genetic distance of 0.08 was found between barcodes of Acanthobrama lisnneri/Acanthobrama telavivensis and Mirogrex terraesanctae, which was a distance relatively small for between-genera and similar to the within-genera distance found between Garra rufa and Garra nana (Heckel, 1843).” )

Response: The sentence was revised to: “A genetic distance of 0.08 was found between barcodes of Acanthobrama lisnneri/Acanthobrama telavivensis and Mirogrex terraesanctae, similar to the within-genus distance found between Garra rufa and Garra nana (Heckel, 1843).

Comment: I don't like the term 'molecular taxonomy' because it refers to a field that do not exist! It's better to use molecular identification or something like this... (Referring to the first sentence in the discussion.)

Response: The term ‘molecular taxonomy’ was changed to ‘molecular identification’. Also, we reviewed the text, found another instance of ‘molecular taxonomy’ in the conclusions section and made a similar change.

Comment: What do you mean with 'anciente origin'? Fishes were introduced? Do you have a reference here? (Referring to the sentence “Cichlids in the Sea of Galilee are of ancient African origin and they constitute a key group in the lake, with both ecological and commercial significance.”).

Response: As the Sea of Galilee is younger than its native species, it is likely that all of them migrated (not introduced) and inhabited the lake. The original meaning in the text was to state that the cichlids were not introduced to the lake, but are native to it and from an African origin. However, to reduce misunderstandings the word ‘ancient’ was removed from the sentence. Additionally, a reference was added (Goren and Ortal, 1999).

Comment: Please provide a reference to “since pure O. niloticus are considered too cold-temperature sensitive to become established there.”

Response: A reference was provided in text (Zengeya et al., 2013)

Reviewer #2

Introduction

Comment: L8: Change “derived pressure” to “derived pressures”.

Response: Text was revised as suggested.

Comment: After “Further human derived pressures on natural fish diversity has also been documented due to introduction and invasion of exotic fish species [10-13].” continue with the following sentence, “Conservation measures intended to mitigate the impact of human derived pressures have largely been slow and inadequate, and as a result, populations of many freshwater species have been declining rapidly [5, 6, 8, 15].”, then end the paragraph with “Assessment of freshwater biodiversity and population status for effective management is, therefore a priority, however, can be considered a great challenge [1, 6, 14, 15].”

Response: Text was revised as suggested.

Comment: Start “Due to its arid climatic conditions ……” as a new paragraph.

Response: Text was revised as suggested.

Comment: L19: Change “about” to “approximately”.

Response: Text was revised as suggested.

Comment: L22-L24: Please change sentence to read as follows: “The native cyprinid Kinneret bleak Mirogrex terraesanctae (Steinitz, 1952), which is endemic to the sea of Galilee, is probably the most abundant species in terms of both numbers and biomass [20].”

Response: Text was revised as suggested.

Comment: L24-L26: Please change sentence to read as follows: “The native cichlid, Galilee tilapia, St. peter’s fish or Mango tilapia Sarotherodon galilaeus (Linnaeus, 1758), which is widespread in central and western Africa [8], is also abundant in terms of numbers and biomass, based on fishery landings, and is considered an important species to local fisheries [25].

Response: Text was revised as suggested.

Comment: L26-L27: Delete the sentence “While M. terraesanctae is endemic, S. galilaeus is widespread in central and western Africa [8]”.

Response: Text was revised as suggested.

Comment: L27-L31: Please change sentence to read as follows: “Non-native species, which were unintentionally introduced, include the European eel Anguilla anguilla (Linnaeus, 1759) that hitchhiked with mullet fry caught in coastal estuaries and stocked in the Sea of Galilee [21], the green swordtail Xiphophorus helleri (Heckel, 1848) [22] and the Peacock bass Cichla kelberi (Kullander and Ferreira, 2006), probably released into the Sea of Galilee by hobbyists [23]”. Out of curiosity, why did you replace X. helleri (spelling is with one “I” at the end, not two “I’s”) with the Oreochromis hybrid? Is it because the Oreochromis hybrid is not an exotic species and this was your focus? Would still be good to include the Oreochromis hybrid example in this paragraph.

Response: Text was revised as suggested. The hybrid tilapia was added back to the text with the appropriate reference to support it.

Comment: L33-L36: Please change sentence to read as follows: “Additionally, significant non-native species including the common carp Cyprinus carpio Linnaeus 1758, the silver carp Hypophthalmichthys molitrix (Valenciennes, 1844), the thinlip grey mullet Chelon ramada (Risso, 1827) and the flathead grey mullet Mugil cephalus Linnaeus, 1758 were introduced to enhance commercial fishing [16, 22].”

Response: Text was revised as suggested.

Comment: L45: rewrite as “[e.g. 26-28].

Response: Text was revised as suggested.

Comment: L59: Replace “Here” with “For this study”.

Response: Text was revised as suggested.

Materials and methods

Comment: L75: Please include the dosage used for anaesthesia.

Response: The anaesthesia dosage was added to the text.

Comment: L77: Please change “100% ethanol” to “99.9% ethanol”.

Response: Text was revised as suggested.

Comment: L77-L78: Please change sentence to read as follows: “In most cases, after identification and sampling, fish recovered from anaesthesia and were returned to the environment after a monitoring period of ……. (please include this)”.

Response: Text was revised as suggested. The monitoring period for the recovery of fish was added.

Comment: L78-L81: Please change sentence to read as follows: “Some fish, however, were sacrificed, photographed and fixed in formalin to be deposited in the National Natural History Collections of the Hebrew University of Jerusalem, Israel with a museum number (Table 1), serving as voucher specimens (transferred to 99.9% ethanol)”.

Response: Text was revised as suggested.

Comment: L79: Why were only some fish photographed and not all fish collected for fin clips? To my understanding, for DNA barcoding protocols, all individuals that were fin clipped should have photographs to be uploaded to the BOLD database. Also, why for so many of your species collected were no individuals taken as voucher specimens/deposited into the National collection? Is it because there are already these species present in the National Collection? I would expect for field sampling records in a study, at least 2 individuals of each species should be accessioned into the Natural collection. This allows other researchers to compare across years and study sites where a particular species is distributed.

Response: Specimens of all sampled species are already present (including collection voucher types) in the NNHC-HUJ. Furthermore, most of the fish were sampled and returned to the environment, including fish that had novel DNA barcodes. Only fish that were not returned were pictured, sampled for DNA extraction and then fixed in formalin and deposited in the collection. 

Ideally, it would be best if for all barcodes in BOLD there would also be voucher specimens deposited in collections for possible future comparative studies. However, restriction on sampling of natural resources and practical considerations, since barcoding is done on very large numbers of specimens, make this ideal situation currently not feasible.

Comments: Table 1:

• Please rephrase table heading to, “Fish species collected, barcoded and deposited at the National Natural History Collections of the Hebrew University of Jerusalem (NNHH-HUJ) by species and location”.

Response: Text was revised as suggested.

• Can you add the columns “Order” and “Family” in Table 1 to allow for the same format in Table 2? 

Response: These columns were added as suggested.

• Can asterisks be added to the hybrids and a cross to A. lissneri as done in Table 2.

Response: Text was revised as suggested.

• Relabel “N samples barcoded” to “N successfully barcoded samples”

Response: Text was revised as suggested.

• Incorrect numbers of successful barcodes are reported in table 1 for A. flaviijosephi (9 instead of 8 for Beit She’an).

Response: We thank the reviewer for thoroughly going over the data and identifying this mismatch. This number was corrected to 9 both in table 1 and in table 2.

• G. rufa (7 instead of 8 for Sea of Galilee).

Response: This was not a mistake, there are 8 G. rufa samples from the Sea of Galilee, one was accidently stored as G. jordanica. This was fixed in the database as per the comment below.

• H. molitrix, H. nobilis, and S. galilaeus when compared to the numbers of samples present in the LKCOX project on the BOLD database. 

Response: Some of the samples that were deposited to BOLD were not analysed in this study since they were either belonging to specimen sampled from aquaculture farms (13 H. molitrix, 7 H. nobilis and 13 H.molitrix x H.nobilis hybrids) or they were not regional (4 S. galilaeus samples from Volta Lake in Ghana). Thus, LKCOX BOLD project contains more samples than reported in the manuscript. However, if one would like to know which are reported in the manuscript and which are not, the sampling location will immediately tell which samples are not from the locations reported in the manuscript.

• For S. galilaeus, in the table you recorded 17 from the Sea of Galilee when in the BOLD database there are 14 records from this site; and for Ein Afek you recorded 4 in the table when there are 2 for this site. Also, on the BOLD database there are other sites including Ginosar Research (4), Naman River (2), and Volta Lake (4), could you have added some of these samples by mistake to the Sea of Galilee and Ein Afek samples?

Response: There are 13 S. galilaeus records from the Sea of Galilee and 4 additional records from a sub-location belonging to the Sea of Galilee (Ginosar Research). For Ein Afek (part of the Naaman River system) there were also two sub-locations (Ein Afek and Naaman River). We prefer to keep the sub – location in the BOLD database, but coordinates were provided so people can see that it’s actually the same ecosystem. 

• Double check the number of 176 published records is correct, in relation to above statement, and explain the difference between this number and 217 published records.

Response: After comparing the records deposited in BOLD with the reported data in tables 1 and 2, we noticed three samples of Carasobarbus canis (cc001-cc003) which were deposited to BOLD but were suspected as contaminants and therefore were not used for the analysis in the paper. To reduce further misunderstandings or possible mistakes, their sequences were removed from the BOLD database.

Without these three C. canis records and without the 37 records detailed in the above comments, after fixing the A. flaviijosephi numbers, the total number of sequences stands on 177 and not 176. This was corrected throughout the text.

We prefer to keep the sub – location in the BOLD database, but coordinates were provided so people can see that it’s actually the same ecosystem. 

• Garra jordanica is not included in the table but is present in the LKCOX project on the BOLD database

Response: This record was mistakenly stored as G. jordanica, it was fixed in the database to have the same identification like the rest of the G. rufa from Israel.

• To my understanding, when depositing vouchers into a National collection, each individual gets a museum accession number. Why is there only one accession number for A. telavivensis when three specimens were deposited? The same for G. rufa, A. flaviijosephi, C. zillii, S. galilaeus and O. aureus. 

Response: In the NNHC-HUJ, each collection event gets an accession number. Meaning that if fish of the same species were collected in the same time and place they all get the same accession number. Sometimes these are sub-divided into HUJXXXXX-01, HUJXXXXX-02 etc. For that reason, there is only one accession number for multiple fish of the same species.

• IS Cyprinus carpio carpio a sub-species of Cyprinus carpio? According to Eschmeyers Catalog of Fishes, there is no Cyprinus carpio carpio listed. Is this meant to be Cyprinus carpio?

Response: This was meant to be Cyprinus carpio, therefore Cyprinus carpio carpio was changed here and in table 2.

Results

Comment: Double check the number of CO1 sequences (176) from the 22 freshwater species, does not add up to the 217 published records on the BOLD database. 

Response: We double checked the numbers as detailed above (in response to comments on table 1), and corrected this number to 177. 

Comment: Table 2: Change “N barcoded samples” to “N successfully barcoded samples”. Check the numbers reported under this heading are correct and correlate to the BOLD database published records.

Response: The text was revised as suggested.

Comment: Figure 2: Can you make GrA a darker purple. Also, can you make GrD a different colour like pink or yellow, currently the blue colour used is too similar to GrC.

Response: GrA, GrC and GrD are haplotypes found for the same species in the same ecosystem (The Sea of Galilee). To make this point, we decided to assign all three of them with different shades of the same color (blue), in contrast to different haplotypes of the same or related species that were labelled by different colors. Therefore, we prefer to leave it in the same color scheme. The figure is also compatible with color blind readers since the haplotype designations are also provided in the figure.

Discussion

Thank you for including some of the limitations to your study as well as a conclusion section. This section brings the findings concisely together.

Comment: Please check that 176 CO1 sequences is correct from the 22 freshwater fish species.

Response: We double checked the numbers as detailed above (in response to comments on table 1), and corrected this number to 177.

Comment: Please rephrase the following to: “Thus, minor frequency new barcodes were identified for a high proportion of the native species, an indication for the genetic divergence and uniqueness of these regional native populations, except for a few discrepancies which will be discussed later on”.

Response: We did not change the text as suggested. This sentence describes the uniqueness of the native populations in the sea of galilee. The next sentence starts with the few discrepancies found between the morphological and the molecular identification and does not belong with the previous sentence.

Comment: Please rephrase the following sentence to: “For 19 out of the 22 species, morphological and DNA barcoding agreed on the species definition”.

Response: We changed the order of this sentence to start with the point that in most cases the morphological and the molecular identification agreed so it now reads as follows: “For 19 out of the 22 species, morphological and DNA barcoding agreed on the species definition, except for a few discrepancies which will be discussed later on”.

---

## [Editor Report · Decision Letter 2]

1 Apr 2022

Establishing and using a genetic database for resolving identification of fish species in the Sea of Galilee, Israel

PONE-D-20-39138R2

Dear Dr. David,

We’re pleased to inform you that your manuscript has been judged scientifically suitable for publication and will be formally accepted for publication once it meets all outstanding technical requirements.

Kind regards,

Mary Peacock, Ph.D.

Academic Editor

PLOS ONE
---

## [Editor Report · Acceptance letter]

5 May 2022

PONE-D-20-39138R2 

Establishing and using a genetic database for resolving identification of fish species in the Sea of Galilee, Israel 

Dear Dr. David:

I'm pleased to inform you that your manuscript has been deemed suitable for publication in PLOS ONE. Congratulations! Your manuscript is now with our production department. 

Kind regards, 

on behalf of

Dr. Mary Peacock 

Academic Editor

PLOS ONE